# CostFilter-AD: Enhancing Anomaly Detection through Matching Cost Filtering

Zhe Zhang [1]   Mingxiu Cai [1]   Hanxiao Wang [2]   Gaochang Wu [† 1]   Tianyou Chai [1]   Xiatian Zhu [† 3]

## Abstract

Unsupervised anomaly detection (UAD) seeks to localize the anomaly mask of an input image with respect to normal samples. Either by reconstructing normal counterparts (reconstruction-based) or by learning an image feature embedding space (embedding-based), existing approaches fundamentally rely on image-level or feature-level matching to derive anomaly scores. Often, such a matching process is inaccurate yet overlooked, leading to sub-optimal detection. To address this issue, we introduce the concept of cost filtering, borrowed from classical matching tasks, such as depth and flow estimation, into the UAD problem. We call this approach *CostFilter-AD*. Specifically, we first construct a matching cost volume between the input and normal samples, comprising two spatial dimensions and one matching dimension that encodes potential matches. To refine this, we propose a cost volume filtering network, guided by the input observation as an attention query across multiple feature layers, which effectively suppresses matching noise while preserving edge structures and capturing subtle anomalies. Designed as a generic post-processing plug-in, CostFilter-AD can be integrated with either reconstruction-based or embedding-based methods. Extensive experiments on MVTec-AD and VisA benchmarks validate the generic benefits of CostFilter-AD for both single- and multi-class UAD tasks. Code and models will be released at https://github.com/ZHE-SAPI/CostFilter-AD.

---

[1]State Key Laboratory of Synthetical Automation for Process Industries, Northeastern University, Shenyang, China. [2]Meta, London, U.K. [3]Surrey Institute for People-Centred Artificial Intelligence, and Centre for Vision, Speech and Signal Processing, University of Surrey, Guildford, U.K. Correspondence to: Gaochang Wu <wugc@mail.neu.edu.cn>, Xiatian Zhu <Eddy.zhuxt@gmail.com>.

*Proceedings of the 42nd International Conference on Machine Learning*, Vancouver, Canada. PMLR 267, 2025. Copyright 2025 by the author(s).

## 1. Introduction

Unsupervised anomaly detection (UAD) plays a critical role in industrial quality inspection (Wu et al., 2024) by identifying anomalies at both image- and pixel-level using models trained solely on normal samples (Zhao, 2023; Chen et al., 2025). Being able to handle the scarcity and diversity of anomalies, as well as address the long-tail distribution of rare anomaly types through anomaly synthesis (Zhao, 2022) have been favored in particular. Existing approaches predominantly follow a "single model per category" paradigm (Liu et al., 2023; Zhang et al., 2023; Chen et al., 2022), while effective, they often result in higher training costs and reduced scalability as anomaly categories expand. Thus, multi-class UAD with a unified model has emerged as a more scalable approach (Lu et al., 2023; He et al., 2024b; Yao et al., 2024). However, the inherent challenges of this task persist due to the diverse characteristics of anomalies across categories, particularly for subtle anomalies around small areas, with low contrast, or proximity to normals, making the unified detection extremely challenging.

Existing UAD methods can be broadly divided into two categories in terms of model design. *Reconstruction*-based methods detect anomalies by comparing residuals or similarities between inputs and their reconstructions. These methods focus on designing networks, such as UNet (Zhao, 2023), Transformer (Lu et al., 2023), and Diffusion (Yao et al., 2024), to address challenges like the "identical shortcut" issue and limited reconstruction capability (You et al., 2022). Such challenges often lead to reconstructions that either retain anomalies in a normal-like style or suffer from undesirable spatial misalignments. On the other hand, *embedding*-based methods (Roth et al., 2022; Damm et al., 2025) operate on the assumption that models trained on normal samples cannot effectively extract features deviating from the normal distribution. This enables these methods to separate anomalous features from normal clusters derived from pre-trained models. Both approaches typically incorporate synthetic anomalies (Zhao, 2022; Zhang et al., 2024; Chen et al., 2025) to simulate real-world anomalies during training, enabling unsupervised learning. In essence, deriving the final anomaly score map in both methods is about matching the input sample with templates (reconstructed or normal training samples) at the image- or feature-level, with anomaly regions exhibiting relatively higher matching costs.

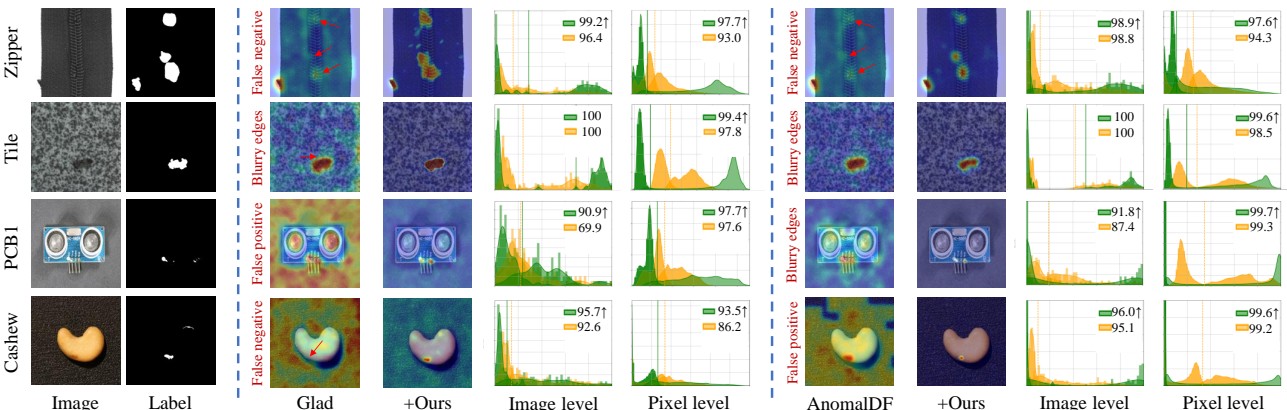

*Figure 1.* Comparison of multi-class UAD results. We present the visualization results and kernel density estimation curves (Parzen, 1962) of image- and pixel-level logits. Baseline results are highlighted in yellow, while ours are shown in green. Our model achieves superior performance by detecting anomalies with less noise and providing a clearer distinction between normal and abnormal logits.

From a matching perspective, we observe that existing UAD methods often prioritize sample reconstruction, precise feature learning, or the utilization of extensive feature banks, while neglecting the intrinsic noise with the matching results. For instance, many approaches directly apply L2 norm (Lu et al., 2023; Li et al., 2024) or cosine similarity (He et al., 2024a;b; Damm et al., 2025; Yao et al., 2024) to compute anomaly score maps. Meanwhile, earlier methods like Draem (Zavrtanik et al., 2021) and JNLD (Zhao, 2022) rely on discriminative networks that implicitly learn the matching process. However, as illustrated by the anomaly localization heatmaps and logits distributions in Fig. 1, such matching noise frequently blurs the boundaries between normal and anomalous regions, rendering simple thresholding of pixel- or image-level logits ineffective. This noise could arise from the unavoidable "identical shortcut" issue or the absence of ideal normal templates, or both (Cao et al., 2024). The significant yet overlooked impact of matching noise would hamper anomaly detection accuracy, especially for subtle or hard-to-detect anomalies.

Inspired by the concept of matching cost filtering (also known as cost volume filtering) from the fields like stereo matching (Hosni et al., 2012; Kendall et al., 2017), depth estimation (Kam et al., 2022), flow estimation (Gudovskiy et al., 2022), and light field rendering (Wu et al., 2025), we reformulate anomaly detection as a three-step paradigm: feature extraction, anomaly cost volume construction, and anomaly cost volume filtering. With this idea, we propose *CostFilter-AD*, a generic post-processing plug-in for enhancing the anomaly detection of both reconstruction-based and embedding-based methods. Conceptually, we introduce a matching cost volume to address "what to match" and a cost volume filtering network to address "how to refine," enabling adaptive noise suppression and more accurate matching between the input image and its corresponding templates.

Specifically, we utilize a pre-trained feature encoder to extract multi-layer features from the input and templates, constructing a multi-layer matching cost volume through global matching across all pixels in the templates. This volume consists of two spatial dimensions, which localize elements in the input, and a matching dimension, which represents the matching scores. To refine this cost volume, we design a filtering network that progressively enhances anomaly detection in a coarse to fine manner. The refinement process leverages the integration of input image features and an initial anomaly map as an attention query, effectively suppressing matching noise while preserving edge structures and capturing subtle anomalies. To further enhance the performance, we expand the matching range of the cost volume by incorporating multiple templates, including reconstructed normal images or normal samples from different views. Additionally, we design a class-aware adaptor that dynamically adjusts the segmentation loss using soft classification logits, prioritizing challenging samples and improving generalization across multiple classes.

Our contributions are as follows: (i) We rethink UAD via matching cost volume filtering to explicitly tackle the intrinsic matching noise – an overlooked yet critical element with existing UAD methods. Under this perspective, we reformulate UAD with a three-step pipeline: feature extraction, matching cost volume construction, and cost volume filtering. (ii) We propose a novel method, CostFilter-AD, characterized by employing multi-layer input observations as attention queries to guide match denoising while preserving edge structures of subtle anomalies. Serving as a general plug-in, it can be seamlessly integrated with both reconstruction-based and embedding-based methods. (iii) Extensive experiments show that our method achieves new state-of-the-art performance under both multi-class and single-class UAD settings on multiple benchmarks.

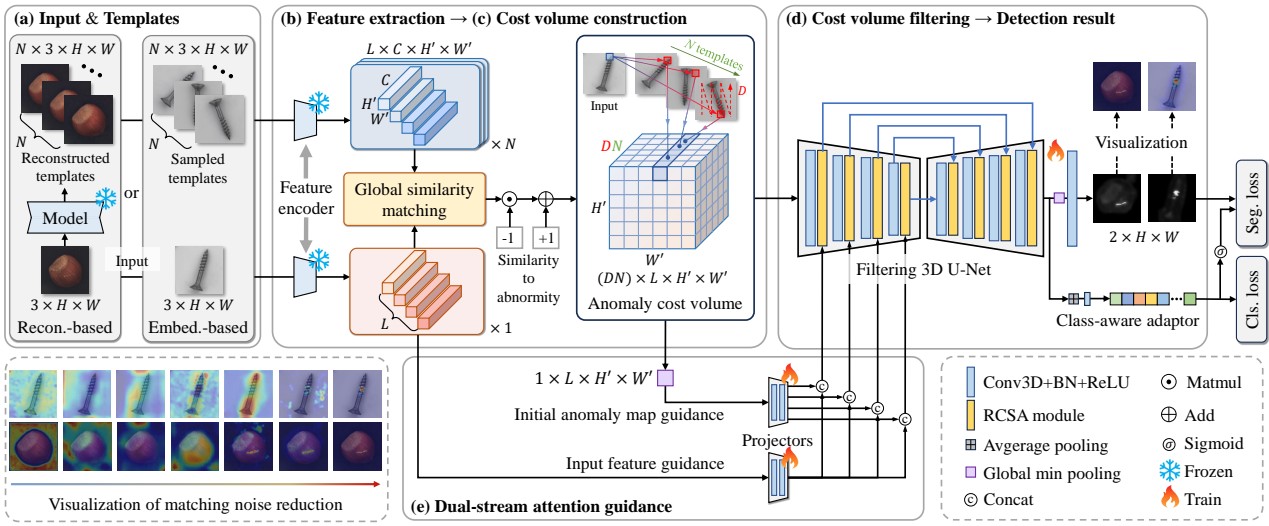

*Figure 2.* Overview of our CostFilter-AD. We reformulate UAD as a matching cost filtering process. (i) First, we employ a pre-trained encoder to extract features from both the input image and the templates (reconstructed normal images or randomly selected normal samples). (ii) Second, we construct an anomaly cost volume based on global similarity matching. (iii) Lastly, we learn a cost volume filtering network, guided by attention queries derived from the input features and an initial anomaly map, to refine the volume and generate the final detection results. (iv) Further, we integrate a class-aware adaptor to tackle class imbalance and enhance the ability to deal with multiple anomaly classes simultaneously.

## 2. Related Work

### 2.1. Unsupervised Anomaly Detection

UAD methods are broadly categorized as embedding-, reconstruction-, and synthesis-based methods (Cao et al., 2024). Embedding-based methods utilize pre-trained models for feature extraction, leveraging techniques like teacher-student networks (Deng & Li, 2022), distribution modeling (Defard et al., 2021; Li et al., 2023; Lee & Choi, 2024), or memory banks (Roth et al., 2022; Bae et al., 2023), but often struggle with adaptability to rare anomalies due to reliance on datasets like ImageNet (Deng et al., 2009). Reconstruction-based methods, including autoencoders (Gong et al., 2019), GANs (Liang et al., 2023; Lv et al., 2024), transformers (You et al., 2022), diffusions (Zhang et al., 2023), and MoE (Meng et al., 2024), aim to rebuild normal patterns, but frequently contend with the "identical shortcut" issue. Synthesis-based methods generate pixel- or feature-level pseudo-anomalies (Zhao, 2022; Chen et al., 2025) to approximate real-world distributions but remain constrained by domain gaps (Zhang et al., 2025; Wang et al., 2023). Additionally, discriminative networks (Zavrtanik et al., 2021; Zhao, 2023) compare image pairs to detect anomalies, yet matching noise remains unresolved.

Recent advancements in diffusion (He et al., 2024b) and foundation models (Caron et al., 2021) have greatly advanced multi-class UAD. For instance, GLAD (Yao et al., 2024) enhances reconstruction with adaptive diffusion steps, while VPDM (Li et al., 2024) minimizes anomaly leakage using vague prototypes. HVQ-Trans (Lu et al., 2023)

enhances feature representation via hierarchical vector quantization, and MambaAD (He et al., 2024a) employs a multi-scale decoder for better reconstruction. Despite these innovations, matching noise remains a critical but overlooked issue. To address this, we propose a novel patch-level matching cost volume filtering method that effectively reduces matching noise and refines anomaly detection, even when reconstructed images or feature embeddings are imperfect.

### 2.2. Cost Volume Filtering in Vision Tasks

Cost volume filtering is a crucial technique in vision tasks, widely used to optimize local matching accuracy (Hosni et al., 2012). In stereo matching, cost volumes correlate left and right image features along the disparity dimension, capturing pixel-level similarities between two views (Kendall et al., 2017; Wang et al., 2024). Similarly, in depth estimation, cost volumes encode multi-view geometric relationships to produce accurate depth maps (Yang et al., 2021; Peng et al., 2022). When extended to motion analysis, optical flow estimation employs cost volumes to represent pixel correspondences across consecutive frames, refining them to improve motion accuracy (Zhang et al., 2021; Garrepalli et al., 2023). These methods employ filtering to process the cost volume, refining matching correspondences and enhancing accuracy (Hosni et al., 2012).

In this paper, we propose CostFilter-AD, a novel approach designed to refine feature matching between input and template images. Unlike prior works that focus exclusively on identifying similar patches for comparison (Lu et al., 2023;

Yao et al., 2024), our method captures anomaly awareness across a diverse range of potential patches. Specifically, CostFilter-AD advances the SOTA approaches in two key ways: (i) by constructing a cost volume for anomaly detection through pixel-wise matching across multiple templates, which enhances detection robustness; and (ii) by employing a cost volume filtering network that leverages multi-layer input observations to guide noise suppression while preserving critical edge information.

## 3. Methodology

We reformulate anomaly detection as a three-step pipeline comprising image feature extraction, anomaly cost volume construction, and anomaly cost volume filtering, as illustrated in Fig. 2. Given the absence of anomalous samples during unsupervised training, we generate synthetic anomalous input images following the protocol outlined in GLAD (Yao et al., 2024). Similar to stereo matching and flow estimation tasks in computer vision, our method matches the input sample $I_{\mathcal{S}} \in \mathbb{R}^{3 \times H \times W}$ (channel, height, and width) with its reconstructed normal image(s) or normal sample(s) from random views, as depicted in Fig. 2 (a). For simplicity, we refer to these reconstructed normal images and normal samples collectively as **Templates** $I_{\mathcal{T}}$. It is important to note that CostFilter-AD is designed to support matching a single input sample with multiple templates.

**Templates for reconstruction-based methods.** Recent advancements have introduced Transformer-based methods (e.g., HQV-Trans (Lu et al., 2023)) and diffusion model-based methods (e.g., GLAD (Yao et al., 2024), DiAD (He et al., 2024b)) to reconstruct high-fidelity normal counterparts for input samples. For Transformer-based reconstruction methods, we set the number of templates $N = 1$.

Diffusion model-based reconstruction methods excel in multi-class UAD by using normal-style images from the final denoising step as templates for feature matching. However, as shown in Fig.1, they often suffer from matching noise (e.g. false negative/positive or blurry edges) due to imperfect reconstructions, reducing detection accuracy (see Fig.6 for analysis). Frequency evolution (Yang et al., 2023) suggests that while the final denoising step is essential for fine-grained detail, it may introduce anomalies through the "identical shortcut." In contrast, intermediate reconstructions, which preserve low-frequency normal information, can offer complementary cues for capturing normal contours.

To this end, during the training phase of CostFilter-AD, we sample $N$ templates from different steps, including the final step, in the backward denoising process to enrich the feature representation. The reconstruction at step $t$ is:

$$I_{t \to 0} = \frac{1}{\sqrt{\bar{\alpha}_t}} \left( I_t - \sqrt{1 - \bar{\alpha}_t}\, \epsilon_\theta(I_t, t) \right), \quad (1)$$

where $\epsilon_\theta$ is the noise predictor of the frozen diffusion model, and $\bar{\alpha}_t$ is manually defined and inversely related to $t$.

**Templates for embedding-based methods.** The primary challenge of embedding-based methods lies in their sensitivity to feature matching noise, which arises from misalignments in size, texture, or views between the input and templates. Existing approaches (Roth et al., 2022; Gu et al., 2023) tackle this problem by leveraging extensive memory banks to search for suitable target templates. Alternatively, we redefine this challenge as a matching noise problem and propose a solution that combines global matching with cost volume filtering. This approach enables us to use only a small number ($N$) of normal images as templates, effectively suppressing matching noise and eliminating the need for large memory banks.

### 3.1. Image Feature Extraction

We utilize the pre-trained DINO model (Caron et al., 2021) to extract image features from the input $I_{\mathcal{S}}$ and templates $I_{\mathcal{T}}$. This process generates a multi-layer input feature tensor $f_{\mathcal{S}} \in \mathbb{R}^{L \times C \times H' \times W'}$ and $N$ multi-layer template feature tensors $f_{\mathcal{T}}$ of the same dimensions, as illustrated in Fig. 2 (b). Here, $L$ represents the number of feature layers, $C$ denotes the feature channels, and $H'$ and $W'$ correspond to the spatial dimensions of the features.

### 3.2. Anomaly Cost Volume Construction

To ensure the generality of CostFilter-AD for both reconstruction-based and embedding-based approaches, we perform global similarity matching across the spatial indices of each template feature. This process is defined as:

$$\mathcal{V}(j, n, l, i) = \frac{f_{\mathcal{S}}^{i,l} \cdot f_{\mathcal{T}}^{n,j,l}}{\|f_{\mathcal{S}}^{i,l}\| \cdot \|f_{\mathcal{T}}^{n,j,l}\|}, \quad (2)$$

where $f_{\mathcal{S}}^{i,l}$ represents the feature vector at the $i$-th spatial index of the input image feature at layer $l \in \{1, 2, \ldots, L\}$, $f_{\mathcal{T}}^{n,j,l}$ denotes the feature vector at the $j$-th spatial index of the $n$-th template feature ($n \in \{1, 2, \ldots, N\}$) at layer $l$, and $\mathcal{V} \in \mathbb{R}^{D \times N \times L \times (H'W')}$ is the resulting similarity volume with $D = H' \times W'$ denoting the matching dimension. Unlike methods that rely on local matching, such as using a single reference image or nearest-neighbor searches within a memory bank, our approach performs global matching across all elements to comprehensively capture feature correlations, as illustrated in Fig. 2 (c).

Since a higher likelihood of anomalies in the input image is indicated by smaller similarity values, we convert the similarity volume into the desired anomaly cost volume $\mathcal{C} \in \mathbb{R}^{D \times N \times L \times (H'W')}$:

$$\mathcal{C}(j, n, l, i) = 1 - \mathcal{V}(j, n, l, i), \quad (3)$$

where larger values indicate a greater probability of anomaly presence. Furthermore, we merge the dimensions $D$ and $N$ into a single dimension since they both correspond to matching results. Additionally, we unfold and reformat $H'W'$ into $H' \times W'$ to represent the spatial dimensions, constructing an anomaly cost volume $\mathcal{C} \in \mathbb{R}^{(DN) \times L \times H' \times W'}$. Additionally, by applying global min pooling along the matching dimension of $\mathcal{C}$, we obtain an initial multi-layer anomaly map $\bar{\mathcal{M}}$, which provides a coarse estimation of anomalies.

### 3.3. Anomaly Cost Volume Filtering

Existing UAD methods often use a Gaussian filter to smooth anomaly score maps (Damm et al., 2025; Yao et al., 2024). However, as shown in Fig.1, these methods tend to produce overly blurred results and fail to eliminate significant background noise. Instead of filtering the final score map, we propose filtering the intermediate anomaly cost volume using a 3D U-Net (Çiçek et al., 2016). This approach effectively reduces matching noise while preserving the edge structures of subtle anomalies.

**Network input.** As illustrated in Fig. 2 (d), the input to our 3D U-Net consists of the constructed anomaly matching cost volume $\mathcal{C} \in \mathbb{R}^{(DN) \times L \times H' \times W'}$, where the matching dimension $DN$ corresponds to the channel dimension of the network, $L$ represents the depth dimension for capturing feature matching across multiple layers, and $H'$ and $W'$ are the spatial dimensions. Additionally, we incorporate the input feature $f_{\mathcal{S}}$ and the initial anomaly map $\bar{\mathcal{M}}$ as guidance for the filtering process.

**Dual-stream attention guidance.** The anomaly cost volume captures extensive global matching information for anomaly detection but is prone to information loss and noise from reconstruction errors or feature misalignment (Fig. 6). To address this, we propose a dual-stream attention guidance mechanism (Fig. 2 (e)). The input image feature $f_{\mathcal{S}}$ provides spatial guidance (SG) to preserve critical details like subtle anomaly edges, while the initial anomaly map $\bar{\mathcal{M}}$ focuses the model's attention on matching dimensions (MG) most likely to detect anomalies. This approach enables the network to capture both global patterns and fine spatial anomaly details effectively.

The dual-stream attention guidance is implemented via the residual channel-spatial attention (RCSA) module, inspired by (Woo et al., 2018) and integrated with residual connections to retain subtle anomaly details, formulated as follows:

$$x_l' = \text{cat}(x_l, h(\bar{\mathcal{M}}), h(f_s^l)),$$

$$x_l^{ca} = \sigma\left(\text{conv}(\text{MP}(x_l')) + \text{conv}(\text{AP}(x_l'))\right) * x_l' + x_l', \quad (4)$$

$$x_l^{sa} = \sigma\left(\text{conv}(\text{cat}(\mu(x_l^{ca}), \max(x_l^{ca})))\right) * x_l^{ca} + x_l^{ca},$$

where $x^l$ denotes the anomaly cost volume feature at layer $l$, processed by the guidance projector $h$ for channel trans-

formation and spatial resolution adjustment. This projector facilitates the concatenation (cat) of dual-stream guidance features with cost volume features along the channel dimension. Additionally, $\sigma$ represents the sigmoid activation, conv denotes 3D convolution, while MP, AP, $\mu$, and max indicate global max pooling, global average pooling, channel-wise mean, and channel-wise max operations, respectively.

The attention-guided features $x_l^{sa}$ are fed into the decoder layer-by-layer via skip connections, with dual-stream attention guidance further optimizing the decoding process. The detailed architecture of our RCSA module is illustrated in Fig. 5. In this way, the proposed dual-stream attention query guidance effectively strengthens global feature matching via residual channel attention and refines pixel-level anomaly localization via residual spatial attention, enabling progressive denoising and precise anomaly detection. More visualizations of the matching noise filtering are shown in Fig. 7 and Fig. 8.

**Class-aware adaptor.** To enhance generalization across multiple anomaly classes, we propose a class-aware adaptor that dynamically guides the segmentation loss with sigmoid-activated soft logits. The adaptor aggregates deep cost volume features via spatial average pooling and projects them onto multi-class classification logits via a fully connected layer, enabling the segmentation head to prioritize challenging samples and adapt to diverse anomaly characteristics.

### 3.4. Anomaly Detection Output Generation

As shown in Fig. 2 (d), following stereo matching (Wang et al., 2024), the filtered anomaly volume undergoes global min pooling along the matching dimension, followed by a convolutional layer and softmax to generate the normal-anomaly score map for localization, denoted as $\mathcal{M} = \text{softmax}(\text{conv}(\min(x)))$. As for detection, the image-level score is the average of the top 250 values of the anomaly score map.

### 3.5. Training and Inference

In this paper, we present our method as a plug-in solution for both reconstruction-based and embedding-based methods. Anomaly volumes are constructed by matching input image features with outputs from frozen reconstruction models or features from randomly selected normal templates.

The matching cost filtering process is designed as a normal-abnormal segmentation task, where the generated anomaly score maps $\mathcal{M}$ are supposed to align with the synthesized anomaly masks $\mathcal{M}_s$. The training objective is defined as:

$$\mathcal{L} = \mathcal{L}_{\text{Focal}}(\mathcal{M}, \mathcal{M}_s, \sigma(\hat{Y}_c)) + \mathcal{L}_{\text{CE}}(\hat{Y}_c, Y)$$
$$+ \alpha \cdot (\mathcal{L}_{\text{Soft-Iou}}(\mathcal{M}, \mathcal{M}_s) + \mathcal{L}_{\text{SSIM}}(\mathcal{M}, \mathcal{M}_s)), \quad (5)$$

where $\mathcal{L}_{\text{Focal}}$ is the focal loss to addresses the normal-

*Table 1.* Multi-class anomaly detection/localization results (image AUROC/pixel AUROC) on MVTec-AD. Models are evaluated across all categories without fine-tuning, with the best results highlighted in bold.

| | Category | PatchCore | OmniAL | DiAD | VPDM | MambaAD | GLAD | GLAD+Ours | HVQ-Trans | HVQ-Trans+Ours | AnomalDF | AnomalDF+Ours |
|---|---|---|---|---|---|---|---|---|---|---|---|---|
| Object | Bottle | **100** / **99.2** | **100** / **99.2** | 99.7 / 98.4 | **100** / 98.6 | **100** / 98.7 | **100** / 98.4 | 99.8 / 97.8 | **100** / 98.3 | **100** / 98.8 | **100** / 87.3 | **100** / 99.1 |
| | Cable | 95.3 / 93.6 | 98.2 / 97.3 | 94.8 / 96.8 | 97.8 / 98.1 | 98.8 / 95.8 | 98.7 / 93.4 | 98.0 / 96.3 | 99.0 / 98.1 | **99.8** / 98.2 | 99.6 / **98.3** | 99.3 / 98.1 |
| | Capsule | 96.8 / 98.0 | 95.2 / 96.9 | 89.0 / 97.1 | **97.0** / 98.8 | 94.4 / 98.4 | 96.5 / 99.1 | 94.3 / **99.2** | 95.4 / 98.8 | 96.4 / 98.9 | 89.7 / 99.1 | 96.1 / **99.2** |
| | Hazelnut | 99.3 / 97.6 | 95.6 / 98.4 | 99.5 / 98.3 | 99.9 / 98.7 | **100** / 99.0 | 97.0 / 98.9 | 99.4 / 99.1 | **100** / 98.8 | **100** / 99.2 | 99.9 / **99.6** | **100** / 99.5 |
| | Metal Nut | 99.1 / 96.3 | 99.2 / 99.1 | 99.1 / 97.3 | 98.9 / 96.0 | 99.9 / 96.7 | 99.9 / 97.3 | **100** / **99.2** | 99.9 / 96.3 | **100** / 97.9 | **100** / 96.7 | **100** / 99.0 |
| | Pill | 86.4 / 90.8 | 97.2 / **98.9** | 95.7 / 95.7 | 97.9 / 96.4 | 97.0 / 97.4 | 94.4 / 97.9 | 97.9 / 97.8 | 95.8 / 97.1 | 96.9 / 96.5 | 97.2 / 98.1 | 98.9 / 98.4 |
| | Screw | 94.2 / 98.9 | 88.0 / 98.0 | 90.7 / 97.9 | 95.5 / 99.3 | 94.7 / 99.5 | 93.4 / **99.6** | 95.4 / **99.6** | **95.6** / 98.9 | 95.3 / 99.0 | 74.3 / 97.6 | 88.5 / 99.0 |
| | Toothbrush | **100** / 98.8 | **100** / 99.0 | 99.7 / 99.0 | 94.6 / 98.8 | 98.3 / 99.0 | **100** / **99.2** | 99.7 / 99.1 | **100** / 98.9 | 99.7 / **99.2** | 99.7 / **99.2** | 99.7 / **99.2** |
| | Transistor | 98.9 / 92.3 | 93.8 / 93.3 | 99.8 / 95.1 | 99.7 / 97.9 | **100** / 97.1 | 99.4 / 90.9 | 99.5 / 91.6 | 99.7 / 99.1 | 99.7 / **99.2** | 96.5 / 95.8 | 97.8 / 97.5 |
| | Zipper | 97.1 / 95.7 | **100** / **99.5** | 95.1 / 96.2 | 99.0 / 98.0 | 99.3 / 98.4 | 96.4 / 93.0 | 99.2 / 97.7 | 97.9 / 97.5 | 98.9 / 98.3 | 98.8 / 94.3 | 98.9 / 96.7 |
| Texture | Carpet | 97.0 / 98.1 | 98.7 / 99.4 | 99.4 / 98.6 | **100** / 98.8 | 99.8 / 99.2 | 97.2 / 98.9 | **100** / 99.1 | 99.9 / 98.7 | **100** / 98.5 | 99.9 / 99.4 | 99.9 / **99.6** |
| | Grid | 91.4 / 98.4 | 99.9 / 99.4 | 98.5 / 96.6 | 98.6 / 98.0 | **100** / 99.2 | 95.1 / 98.1 | **100** / **99.5** | 97.0 / 97.0 | 99.3 / 98.3 | 98.2 / 97.8 | **100** / **99.5** |
| | Leather | **100** / 99.2 | 99.0 / 99.3 | 99.8 / 98.8 | **100** / 99.2 | **100** / 99.4 | 99.5 / **99.7** | **100** / 99.6 | **100** / 98.8 | **100** / 99.3 | **100** / **99.7** | **100** / **99.7** |
| | Tile | 96.0 / 90.3 | 99.6 / 99.0 | 96.8 / 92.4 | **100** / 94.5 | 98.2 / 93.8 | **100** / 97.8 | **100** / 99.4 | 99.2 / 92.2 | **100** / 95.0 | **100** / 98.5 | **100** / **99.6** |
| | Wood | 93.8 / 90.8 | 93.2 / 97.4 | **99.7** / 93.3 | 98.2 / 95.3 | 98.8 / 94.4 | 95.4 / 96.8 | 97.4 / 97.4 | 97.2 / 92.4 | 98.5 / 94.3 | 97.9 / 97.6 | 98.9 / **98.2** |
| | Mean | 96.4 / 95.7 | 97.2 / 98.3 | 97.2 / 96.8 | 98.4 / 97.8 | 98.6 / 97.7 | 97.5 / 97.3 | 98.7 / 98.2 | 98.0 / 97.3 | **99.0** / 98.0 | 96.8 / 98.1 | 98.5 / **98.8** |

anomaly imbalance, $\mathcal{L}_{\text{Soft-Iou}}$ refines anomaly region localization, $\mathcal{L}_{\text{SSIM}}$ preserves structural consistency, and $\mathcal{L}_{\text{CE}}$ enhances multi-class classification. Notably, the parameter $\gamma$ in $\mathcal{L}_{\text{Focal}}$ is adjusted as $\gamma = \gamma_0 - \sigma(\hat{Y}_c)$ when the adaptor classifies correctly, and $\gamma = \gamma_0$ otherwise. Thus, the class-aware adaptor modulates $\gamma$ via its predicted logits $\hat{Y}_c$ to enhance multi-class segmentation.

The inference process similarly constructs and filters the matching cost volume, yielding anomaly map $\mathcal{M}$. To integrate with the baseline response, we compute a weighted sum $\lambda \cdot \mathcal{M} + (1 - \lambda) \cdot \mathcal{M}_{\text{baseline}}$ for anomaly localization and detection, where $\lambda \in [0, 1]$ compensates for potential scale differences between the two components.

## 4. Experimental Evaluation

**Datasets.** (1) MVTec-AD (Bergmann et al., 2019) is a widely used dataset for evaluating UAD models in industrial applications. It comprises 5,324 high-resolution images across 10 objects and 5 texture categories. Each category provides anomaly-free training samples and test images with various surface defects. (2) VisA (Zou et al., 2022) is a large-scale dataset comprising 10,821 high-resolution images spanning 12 subsets, including 9,621 normal and 1,200 anomalous images. It covers a wide range of surface defects (e.g., dents, scratches, color spots, and cracks) as well as structural anomalies such as misalignment and missing components. (3) MPDD (Jezek et al., 2021) is a challenging dataset consisting of 1,346 images from 6 metal part categories. It includes 888 normal training samples and 458 test images, with pixel-level annotations for anomaly regions. (4) BTAD (Mishra et al., 2021) comprises 2,830 images from three industrial categories for UAD.

**Evaluation metrics.** For a comprehensive evaluation, we adopt widely used metrics following prior works. For

anomaly detection, we report image-level AUROC (I-AUROC), AUPRC (I-AP), and F1max (I-F1max). For anomaly localization, we employ pixel-level AUROC (P-AUROC), AUPRC (P-AP), F1max (P-F1max), and Area Under the Per-Region-Overlap Curve (AUPRO). In the main text, we primarily report image/pixel AUROC, while other metrics are detailed in the appendix.

**Implementation details.** To validate our method as a plug-in solution, we integrate it with three recent multi-class UAD methods: GLAD (Yao et al., 2024) (diffusion-based), HVQ-Trans (Lu et al., 2023) (transformer-based) and AnomalDF (Damm et al., 2025) (embedding-based) under the full-shot setting, with full access to the training data set. We adhere to the original settings for a fair comparison. (i) Input images are resized to $256 \times 256$ for GLAD and AnomalDF, and $224 \times 224$ for HVQ-Trans. (ii) For the first two baselines, three templates ($N = 3$) are randomly selected either from 25 diffusion denoising steps or the same-category training set. For HVQ-Trans, $N = 1$ as it does not reconstruct intermediates. (iii) In our setting, AnomalDF (+Ours) refers to a variant of AnomalyDINO that dynamically samples templates from the full training set for each input, in contrast to the original fixed few-shot protocol with a static template pool, thereby achieving a trade-off between template diversity and memory efficiency. (iv) Multi-layer features ($L = 4$) are extracted using a pre-trained DINO (Caron et al., 2021) with ViT-B/8 for GLAD and AnomalDF, and a pre-trained EfficientNet-B4 (Tan & Le, 2021) for HVQ-Trans. (v) To reduce memory usage, the anomaly cost volume is trimmed to the top $D$ smallest channels as input for the first two baselines, aligning with HVQ-Trans, where the volume inherently has $D$ channels with a single template. (vi) We train our models from scratch for 40 epochs with a batch size of 8, using the Adam optimizer initialized with a learning rate of $1 \times 10^{-3}$. (vii) The loss weight $\alpha$ is set to 0.1 by default.

*Table 2.* Multi-class anomaly detection/localization results (image AUROC/pixel AUROC) on VisA. Models are evaluated across all categories without fine-tuning, with the best results highlighted in bold.

| Category | | JNLD | OmniAL | DiAD | VPDM | MambaAD | GLAD | GLAD+Ours | HVQ-Trans | HVQ-Trans+Ours | AnomalDF | AnomalDF+Ours |
|---|---|---|---|---|---|---|---|---|---|---|---|---|
| Complex Structure | PCB1 | 82.9 / 98.9 | 77.7 / 97.6 | 88.1 / 98.7 | **98.2** / 99.6 | 95.4 / **99.8** | 69.9 / 97.6 | 90.9 / 97.7 | 95.1 / 99.5 | 96.3 / 99.3 | 87.4 / 99.3 | 91.8 / 99.7 |
| | PCB2 | 79.1 / 95.0 | 81.0 / 93.9 | 91.4 / 95.2 | **97.5** / 98.8 | 94.2 / **98.9** | 89.9 / 97.1 | 93.2 / 95.7 | 93.4 / 98.1 | 97.0 / 98.0 | 81.9 / 94.2 | 95.7 / 98.0 |
| | PCB3 | 90.1 / 98.5 | 88.1 / 94.7 | 86.2 / 96.7 | **94.5** / 98.7 | 93.7 / **99.1** | 93.3 / 96.2 | 90.5 / 97.4 | 88.5 / 98.2 | 89.8 / 97.7 | 87.4 / 96.5 | 94.0 / 98.9 |
| | PCB4 | 96.2 / 97.5 | 95.3 / 97.1 | 99.6 / 97.0 | **99.9** / 97.8 | **99.9** / 98.6 | 99.0 / **99.4** | 99.4 / 99.3 | 99.3 / 98.1 | 98.7 / 97.8 | 96.7 / 97.3 | 98.1 / 98.9 |
| Multiple Instances | Macaroni1 | 90.5 / 93.3 | 92.6 / 98.6 | 85.7 / 94.1 | **97.5** / 99.6 | 91.6 / 99.5 | 93.1 / **99.9** | 96.0 / 99.9 | 88.7 / 99.1 | 93.7 / 99.4 | 88.0 / 98.2 | 95.3 / **99.9** |
| | Macaroni2 | 71.3 / 92.1 | 75.2 / 97.9 | 62.5 / 93.6 | 85.7 / 99.0 | 81.6 / 99.5 | 74.5 / 99.5 | 79.7 / 99.6 | 84.6 / 98.1 | **88.3** / 98.5 | 75.9 / 96.9 | 82.2 / **99.7** |
| | Capsules | 91.4 / **99.6** | 90.6 / 99.4 | 58.2 / 97.3 | 79.5 / 99.1 | 91.8 / 99.1 | 88.8 / 99.3 | 89.1 / 99.0 | 74.8 / 98.4 | 80.1 / 97.6 | **93.6** / 97.0 | 88.5 / 98.6 |
| | Candles | 85.4 / 94.5 | 86.8 / 95.8 | 92.8 / 97.3 | 97.2 / 99.4 | 96.8 / 99.0 | 86.4 / 98.8 | 90.5 / 98.8 | 95.6 / 99.1 | **97.8** / 99.2 | 90.3 / 96.1 | 95.1 / **99.4** |
| Single Instance | Cashew | 82.5 / 94.1 | 88.6 / 95.0 | 91.5 / 90.9 | 90.0 / 98.0 | 94.5 / 94.3 | 92.6 / 86.2 | 95.7 / 93.5 | 92.2 / 98.7 | 94.1 / 99.3 | 95.1 / 99.2 | **96.0** / 99.6 |
| | Chewing gum | 96.0 / 98.9 | 96.4 / 99.0 | 99.1 / 94.7 | 99.0 / 98.6 | 97.7 / 98.1 | 98.0 / 99.6 | **99.4** / 99.7 | 99.1 / 98.1 | 99.3 / 99.5 | 98.0 / 99.3 | 99.1 / **99.7** |
| | Fryum | 91.9 / 90.0 | 94.6 / 92.1 | 89.8 / 97.6 | 92.0 / **98.6** | 95.2 / 96.9 | 97.2 / 96.8 | **97.7** / 97.3 | 87.1 / 97.7 | 88.9 / 97.8 | 93.4 / 96.1 | 96.9 / 97.9 |
| | Pipe Fryum | 87.5 / 92.5 | 86.1 / 98.2 | 96.2 / 97.6 | 98.8 / 99.4 | 98.7 / 99.1 | 98.0 / 98.9 | 95.8 / 99.3 | 97.5 / 99.4 | 96.6 / 99.5 | 98.0 / 99.1 | **99.1** / **99.7** |
| | Mean | 87.1 / 95.2 | 87.8 / 96.6 | 86.8 / 96.0 | 94.2 / 98.9 | **94.3** / 98.5 | 90.1 / 97.4 | 93.2 / 98.1 | 91.3 / 98.5 | 93.4 / 98.6 | 90.5 / 97.5 | **94.3** / 99.2 |

*Table 3.* Multi-class UAD evaluation across additional baselines and benchmarks using seven metrics, reporting category-wise mean results for each benchmark.

| Benchmark | Method | Image-level | | | Pixel-level | | | |
|---|---|---|---|---|---|---|---|---|
| | | AU-ROC | AP | F1max | AU-ROC | AP | F1max | AUPRO |
| MVTec-AD | UniAD | 97.5 | 99.1 | 97.0 | 96.9 | 44.5 | 50.5 | 90.6 |
| | +Ours | 99.0 | 99.7 | 98.1 | 97.5 | 60.5 | 59.9 | 91.3 |
| | Dinomaly | 99.6 | **99.8** | 99.0 | 98.3 | 68.7 | 68.7 | 94.6 |
| | +Ours | **99.7** | **99.8** | **99.1** | **98.4** | **68.9** | **68.9** | **94.8** |
| VisA | UniAD | 91.5 | 93.6 | 88.5 | 98.0 | 32.7 | 38.4 | 76.1 |
| | +Ours | 92.1 | 94.0 | 88.9 | 98.6 | 34.0 | 39.0 | 86.4 |
| | Dinomaly | **98.7** | 98.9 | 96.1 | 98.7 | 52.5 | 55.4 | 94.5 |
| | +Ours | **98.7** | **99.0** | **96.3** | **98.8** | **53.2** | **55.8** | **94.7** |
| MPDD | HVQ-Trans | 86.5 | 87.9 | 85.6 | 96.9 | 26.4 | 30.5 | 88.0 |
| | +Ours | 93.1 | 95.4 | 90.3 | 97.5 | 34.1 | 37.0 | 82.9 |
| | Dinomaly | 97.3 | **98.5** | 95.6 | 99.1 | 60.0 | 59.8 | **96.7** |
| | +Ours | **97.5** | **98.5** | **95.8** | **99.2** | **60.2** | **59.9** | **96.7** |
| BTAD | HVQ-Trans | 90.9 | 97.8 | 94.8 | 96.7 | 43.2 | 48.7 | 75.6 |
| | +Ours | 93.3 | **98.6** | **96.0** | 97.3 | 47.0 | 50.2 | 76.2 |
| | Dinomaly | 95.4 | 98.5 | 95.5 | 97.9 | 70.1 | 68.0 | 76.5 |
| | +Ours | **95.5** | **98.6** | 95.8 | **98.1** | **74.3** | **69.8** | **77.5** |

*Table 4.* Extended studies on single-class UAD with our models.

| Benchmark | Method | Image-level | | | Pixel-level | | | |
|---|---|---|---|---|---|---|---|---|
| | | AU-ROC | AP | F1max | AU-ROC | AP | F1max | AUPRO |
| MVTec-AD | Glad | 99.0 | 99.7 | 98.2 | 98.7 | 63.8 | 63.7 | 95.2 |
| | +Ours | **99.3** | **99.7** | **98.3** | **98.9** | **66.2** | **65.0** | **96.4** |
| VisA | Glad | 99.3 | 99.6 | 97.6 | 98.3 | 35.8 | 42.4 | 94.1 |
| | +Ours | **99.5** | **99.7** | **98.1** | **98.6** | **37.3** | **45.3** | **94.5** |

method surpasses baselines by 4.9%/1.4%, 2.3%/1.3%, and 1.8%/1.7%, respectively. We attribute this to the method's effective mitigation of matching noise, a critical yet often overlooked challenge that broadly exists in reconstruction- and embedding-based methods. These results highlight its strong generalization and superior performance across diverse anomaly classes.

**Multi-class UAD on VisA.** Due to its complex anomaly types and diversity of data distribution, VisA poses more challenges in terms of anomaly detection. Table 2 enumerates the results comparison of our methods and recent methods. Specifically, we outperform the three baselines by 3.1%/0.7%, 2.1%/0.1%, and 3.8%/1.7%, respectively, with more metrics that can be referred to in Table 13 and Table 16. These results also validate the effectiveness, generalizability, and robustness of our proposed method.

**Multi-class UAD across more baselines and benchmarks.** To further assess the generalizability of our plug-in design, we evaluate CostFilter-AD on a broader set of recent multi-class UAD baselines and benchmarks, using seven comprehensive metrics at both image and pixel levels. As summarized in Table 3, we present category-wise mean results for each benchmark to provide an overall performance perspective; detailed per-class metrics are available in Table 9 to Table 20. Our method consistently improves performance across all baselines, datasets, and evaluation dimensions. For example, when integrated with Dinomaly (Guo et al., 2025), CostFilter-AD achieves AUROC scores of 99.6%/98.5% (image/pixel) on MVTec-AD, 98.8%/98.8%

## 4.1. Quantitative Comparison

We reproduced the results of UniAD (You et al., 2022), GLAD, HVQ-Trans, AnomalDF, and Dinomaly (Guo et al., 2025), and integrated our method into these multi-class UAD baselines to validate its effectiveness. Moreover, we conducted a comprehensive comparison with various advanced methods, including synthetic-based JNLD (Zhao, 2022), CNN-based OmniAL (Zhao, 2023), diffusion-based DiAD (He et al., 2024b) and VPDM (Li et al., 2024), and Mamba-based MambaAD (He et al., 2024a).

**Multi-class UAD on MVTec-AD.** As shown in Table 1, integrating our method yields significant AUROC improvements of 1.2%/0.9% for GLAD, 1.0%/0.7% for HVQ-Trans, and 1.7%/0.7% for AnomalDF in image- and pixel-level detection, achieving state-of-the-art results. More metrics in Table 9 to Table 12 further validate its effectiveness. Notably, in texture anomalies like the grid category, our

*Table 5.* Ablation studies of Glad+Ours on MVTec-AD. "$DN \rightarrow$ depth/channel" refers to mapping the matching dimension into the depth/channel dimension of the 3D U-Net. $\mathcal{C}_0$ denotes the volume uisng the final denoising step, $\mathcal{C}_{N-1}$ indicates uisng $N-1$ intermediate steps. SG and MG denote dual-stream attention guidance. $\mathcal{L}_F$ is focal loss, $\mathcal{L}_{CE}$ corresponds to the class-aware adaptor, and $\mathcal{L}_S$ is the combination of $\mathcal{L}_{SSIM}$ and $\mathcal{L}_{Soft-Iou}$.

| $DN \rightarrow$ depth | $DN \rightarrow$ channel | | | | $\mathcal{L}_F$ | $\mathcal{L}_{CE}$ | $\mathcal{L}_S$ | Results |
| | $\mathcal{C}_0$ | $\mathcal{C}_{N-1}$ | SG | MG | | | | |
|---|---|---|---|---|---|---|---|---|
| ✓ | - | - | - | - | ✓ | - | - | 87.8/89.0 |
| - | ✓ | - | - | - | ✓ | - | - | 96.2/96.8 |
| - | ✓ | ✓ | - | - | ✓ | - | - | 96.7/97.3 |
| - | ✓ | ✓ | ✓ | - | ✓ | - | - | 97.8/97.5 |
| - | ✓ | ✓ | ✓ | ✓ | ✓ | - | - | 98.3/97.8 |
| - | ✓ | ✓ | ✓ | ✓ | ✓ | ✓ | - | 98.5/98.0 |
| - | ✓ | - | ✓ | ✓ | ✓ | ✓ | ✓ | 98.4/97.6 |
| - | ✓ | ✓ | ✓ | ✓ | ✓ | ✓ | ✓ | **98.7/98.2** |

on VisA, 97.4%/99.1% on MPDD, and 95.8%/98.1% on BTAD. We attribute these consistent gains to its ability to effectively filter matching noise that is widespread yet often overlooked, while preserving subtle anomaly structures and features across a wide range of anomaly categories.

**Single-class UAD using our unified multi-class model.** We further explored the "single model per category" paradigm by applying our unified multi-class model to filter anomaly volumes generated by category-specific diffusion models in GLAD (Yao et al., 2024). Notably, GLAD also offers single-class diffusion models with an image resolution of $256 \times 256$, which were utilized in this analysis. For simplicity, we employ our unified model without additional fine-tuning, and without training a separate model for each class. As shown in Table 4, our method consistently enhances class-wise mean performance in both image-level detection and pixel-level localization across two benchmarks.

### 4.2. Qualitative Comparison

We conducted qualitative experiments on the MVTec-AD and VisA benchmarks to assess the performance of our method against existing baselines. As illustrated in Fig. 3, existing methods often struggle with significant matching noise, impairing anomaly localization accuracy. In contrast, our method effectively mitigates this issue, enabling precise anomaly segmentation. Additionally, the appendix provides supplementary visualizations, including progressive noise reduction, comparative anomaly localization results, and kernel density estimations for image- and pixel-level logits.

### 4.3. Ablation Studies on Each Component and Losses

To evaluate the contributions of each component and loss term in our method, we conducted ablation studies on the MVTec-AD benchmark using the Glad+Ours setting. Table 5 highlights the following key findings: (i) When di-

*Table 6.* Evaluation of our models on various anomaly volumes.

| Test ╲ Train | MVTec-AD | | VisA | |
| | Recon. | Embed. | Recon. | Embed. |
|---|---|---|---|---|
| Recon. | 98.7 / **98.2** | 97.5↓ / 97.1↓ | **93.2** / 98.1 | 92.6↓ / 98.0↓ |
| Embed. | 94.5↓ / 98.0↓ | 98.5 / 98.8 | 85.6↓ / 96.9↓ | **94.3** / 99.2 |
| Hybrid | **98.8**↑ / 98.1 | **98.6**↑ / **98.9**↑ | 93.1 / **98.2**↑ | 92.9 / **99.3**↑ |

rectly adhering to stereo matching methods designed for local pixel-level matching, results drop to 87.8%/89.0%, likely due to feature contamination caused by global matching across multiple templates. In contrast, mapping the matching correspondence into the channel dimension boosts detection performance effectively. (ii) Using $\mathcal{C}_0$, which relies on the final denoised image as a template, achieves 96.2%/96.8%, while incorporating $N-1$ randomly selected intermediate denoised images further improves performance by 0.5%/0.5%. (iii) The proposed dual-stream attention guidance mechanism significantly enhances the filtering network's learning. SG boosts spatial anomaly attention to 97.8%/97.5%, while the initial anomaly map-guided attention (MG) further improves channel matching, achieving 98.3%/97.8%. (iv) Loss functions: Focal loss serves as a fundamental loss, while $\mathcal{L}_{CE}$ improves performance to 98.5%/98.0%. Structural similarity and soft-IoU losses enhance structural consistency, and joint optimization yields a peak performance of 98.7%/98.2%. Overall, these findings underscore the complementary roles of each component and loss function in enhancing anomaly detection and localization, ensuring robustness and precision.

### 4.4. Further Analysis

**Compatibility exploration with different template types.** Given that the templates for anomaly cost volume construction in this paper can originate from either multi-step reconstructions or randomly selected normal images, we further investigated the compatibility of our method across these different templates. Table 6 reveals an inevitable performance decline when our embedding-based model (Embed, i.e., AnomalDF+Ours) processes reconstruction-based anomaly volumes (Recon, from Glad), and vice versa, due to distinct feature distribution differences between two anomaly volume types, as analyzed in Fig. 1. To address this, we trained a unified model (Hybrid) using anomaly volumes alternately sourced from both template types. The results show that the hybrid model matches or even surpasses the performance of our models tailored to individual types, highlighting the robustness and compatibility of our method to different template types across multi-class anomalies.

**Time and memory efficiency.** Table 7 presents a comparative analysis of computational efficiency across multiple baselines, reporting parameter count, FLOPs, memory con-

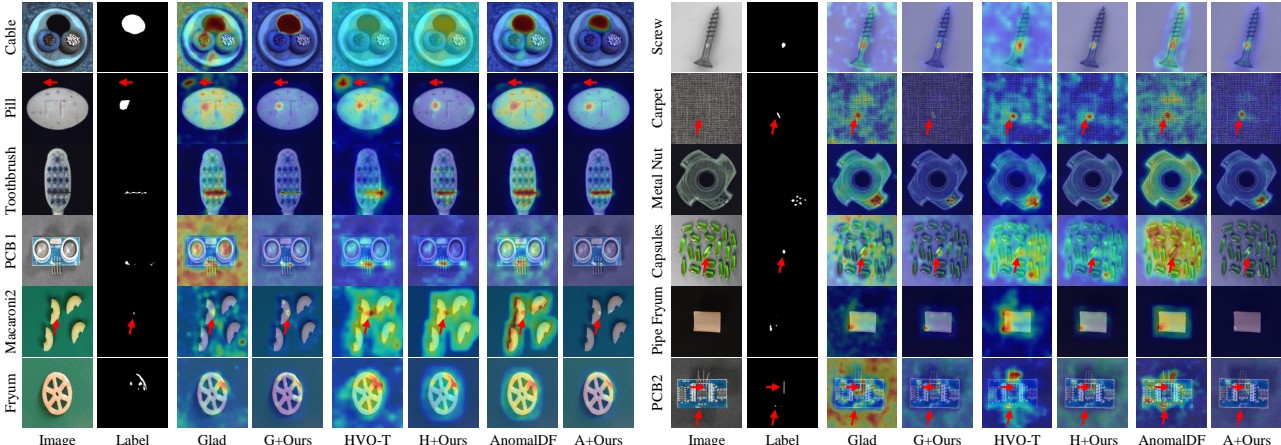

*Figure 3.* Qualitative comparison of multi-class anomaly localization between our method and GLAD (G), HVQ-Trans (H), and AnomalDF (A) on MVTec-AD (top 3 rows) and VisA (bottom 3 rows). By integrating with existing methods, our approach effectively mitigates matching noise (e.g., false negatives in PCB2, false positives in Pill, and blurred boundaries in Carpet), enhancing anomaly detection.

*Table 7.* Comprehensive comparison of baselines and + Ours in terms of parameter size (#Params), computational cost (FLOPs), memory usage (Mem.), and per-image inference time (Inf.).

| Method | #Params | FLOPs | Mem. (GB) | Inf. (s/image) |
|---|---|---|---|---|
| UniAD / +Ours | 7.7M / +43.0M | 198.0G / +26.0G | 4.53 / +0.56 | 0.01 / +0.04 |
| Glad/+Ours | 1.3B / +43.8M | >2.2T / +32.7G | 8.79 / +2.07 | 3.96 / +0.37 |
| HVQ-Trans/+Ours | 18.0M / +43.0M | 7.4G / +26.0G | 4.78 / +0.94 | 0.05 / +0.07 |
| AnomalDF/+Ours | 21.0M / +43.8M | 4.9G / +32.7G | 3.25 / +0.82 | 0.31 / +0.32 |
| Dinomaly/+Ours | 132.8M / +43.6M | 104.7G / +14.3G | 4.32 / +1.11 | 0.11 / +0.05 |

sumption, and per-image inference latency. All measurements were obtained with a batch size of 1 on an A100 GPU (40 GB), both with and without our method. The results demonstrate that our plug-in consistently improves performance while introducing only reasonable memory overhead and modest computational cost. Notably, the increase in memory usage is minimal, and the additional inference latency remains low, particularly when compared to diffusion-based approaches. Minor variations in #Params stem from projecting matching features with different channel dimensions (e.g., 196, 768, 1024) into a unified 96-dimensional space. By maintaining architectural compatibility and low integration overhead, our method enables efficient and accurate anomaly detection and localization, reinforcing its applicability to real-world deployment scenarios.

### 4.5. Failure Case Analysis

As illustrated in Fig. 4, we present representative failure cases from six categories in MVTec-AD and VisA, demonstrating the results of our method before and after filtering. While our method effectively suppresses the matching noise, it mainly relies on the presence of anomaly-relevant signals in the cost volume. To address situations where such signals are limited due to low-resolution inputs or insufficient feature extraction, we introduce input image features for spatial guidance. Nevertheless, recovery remains challenging in

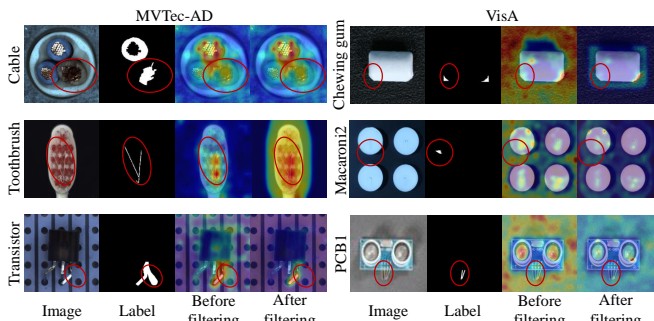

*Figure 4.* Failure cases on MVTec-AD and VisA. Some subtle anomalies may lead to inaccurate or missed localization, potentially due to limited representation in the constructed cost volume.

these cases, indicating a promising direction for future work.

## 5. Conclusion

We propose *CostFilter-AD* as a generic post-processing plug-in for both reconstruction and embedding, to address the matching noise in unsupervised anomaly detection (UAD), a critical challenge hindering precise anomaly detection. By proposing anomaly cost volume filtering, we reformulate UAD as a three-step pipeline: feature extraction, anomaly cost volume construction, and cost volume filtering, which effectively mitigates matching noise caused by the "identical shortcut" issue and the misalignment noise. Concurrently, we propose a dual-stream attention guidance that enhances global feature matching and spatial anomaly refinement via a residual channel-spatial attention module, enabling progressive denoising and accurate detection. Extensive experiments on MVTec-AD and VisA benchmarks show that CostFilter-AD achieves significant improvements with minimal computational overhead, excelling in both multi-class and single-class UAD tasks. This work offers a scalable and precise solution for anomaly detection, underscoring its practical value for real-world applications.

## Acknowledgements

This work is supported in part by the Research Program of the Liaoning Liaohe Laboratory under Grant No. LLL23ZZ-05-01, in part by the Natural Science Foundation of Liaoning Province of China under Grant No. 2024-MSBA-42, in part by the Key Research and Development Program of Liaoning Province under Grant No. 2023JH26/10200011, in part by the Science and Technology Major Project of Liaoning Province under Grant No. 2024JH1/11700048, in part by the Fundamental Research Funds for the Central Universities under Grant No. N25YJS002, in part by the Major Program of the National Natural Science Foundation of China (NSFC) under Grant No. 61991404, and in part by the Program of China Scholarship Council 202306080142.

## Impact Statement

This paper presents work whose goal is to advance unsupervised anomaly detection through *CostFilter-AD*, a generic post-processing plug-in designed to address matching noise, which is widely present in existing methods. We reformulate anomaly detection as a three-step pipeline: feature extraction, cost volume construction, and cost volume filtering, providing a scalable and precise solution. There are many potential societal consequences of our work, none of which we feel must be specifically highlighted here.

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

## Appendix Overview

This appendix provides supplementary insights supporting the main manuscript, organized as follows:

— **Sec. A** provides more implementation details.

— **Sec. B** visualizes challenges in diffusion-based reconstruction and advancements enabled by our method.

— **Sec. C** visualizes the progressive reduction of matching noise by our CostFilter-AD model.

— **Sec. D** provides evaluations of CostFilter-AD under multiple original AnomalyDINO full-shot settings.

— **Sec. E** showcases comprehensive qualitative results for each category on the MVTec-AD dataset.

— **Sec. F** showcases comprehensive qualitative results for each category on the VisA dataset.

— **Sec. G** reports comprehensive per-category quantitative results on the MVTec-AD dataset.

— **Sec. H** reports comprehensive per-category quantitative results on the VisA dataset.

— **Sec. I** reports comprehensive per-category quantitative results on the BTAD dataset.

— **Sec. J** reports comprehensive per-category quantitative results on the MPDD dataset.

## A. More Implementation Details.

(i) **AnomalDF/AnomalDF+Ours**. In this paper, we refer to the full-shot version of AnomalDINO (Damm et al., 2025) as AnomalDF, where "F" denotes the full-shot setting. AnomalDINO explores two experimental settings: few-shot and full-shot. The few-shot setting assumes access to only a few fixed normal templates per category, whereas the full-shot setting constructs a big memory bank of all normal template features from the training set. However, this full-shot approach incurs more storage overhead.

To address this, we optimize global feature matching and denoising by leveraging a limited selection of normal templates. **During training**, AnomalDF (+Ours) uses a fixed number ($N = 3$) of templates per input image, randomly sampled from the full training set for each input, rather than drawn from a fixed template set as in the original few-shot setting of AnomalyDINO. Our dynamic sampling ensures the template pool covers the full training distribution; thus, we classify it as full-shot, offering a trade-off between template diversity and memory efficiency. **During testing**, for fairness, we evaluate AnomalDF (+Ours) using our dynamic 3-shot sampling protocol, as reflected in the results reported in our submission.

(ii) **Training setup.** We train with a batch size of $B = 8$ and employ a ReduceLROnPlateau scheduler, which reduces the learning rate by half when the loss stagnates, ensuring stable convergence. For Eq. 5, the class-aware adaptor is configured with $\gamma_0 = 3$. For a fair comparison, we adopt the baseline configurations. Specifically, for GLAD and AnomalDF, we extract features from the 3rd, 6th, 9th, and 12th layers of a pre-trained DINO model (Caron et al., 2021), while for HVQ-Trans, we leverage features from the 1st, 5th, 9th, and 21st layers of pre-trained EfficientNet-B4 (Tan & Le, 2021). In the fourth decoder layer of the filtering network, a 3D convolution along the depth dimension condenses the matching cost volume from $L = 4$ to 1. The resulting feature representation is then used to generate the normal-anomaly score map in Section 3.4.

Additionally, the $\mathcal{M}$ used in the loss function (Eq. 5) represents the anomaly score map. Moreover, $\mathcal{L}_{\text{Focal}}$ refers to Focal Loss (Lin et al., 2017), where the parameter $\gamma$ modulates the model's emphasis on hard-to-classify samples. $\mathcal{L}_{\text{Soft-Iou}}$ represents Soft Intersection-over-Union Loss (Rahman & Wang, 2016), refining anomaly localization through IoU optimization. $\mathcal{L}_{\text{SSIM}}$ corresponds to Structural Similarity Index Loss (Wang et al., 2004), ensuring spatial structural consistency. Lastly, $\mathcal{L}_{\text{CE}}$ denotes Cross-Entropy Loss (Krizhevsky et al., 2012), enhancing multi-class classification by mitigating entropy-based uncertainty.

(iii) **Evaluation protocol**. We adopt standard practice for qualitative visualization by plotting the abnormal map logits of a single image to ensure clearer comparisons. For quantitative evaluation and KDE logit curve generation, we normalize the normal-abnormal map logits across all images within a category, improving detection robustness and comparability.

(iv) **Utilization of input/reconstructed features in reconstruction-based methods.** For reconstruction-based UAD methods such as HVQ-Trans, UniAD, and Dinomaly, we construct the cost volume directly using both the input and

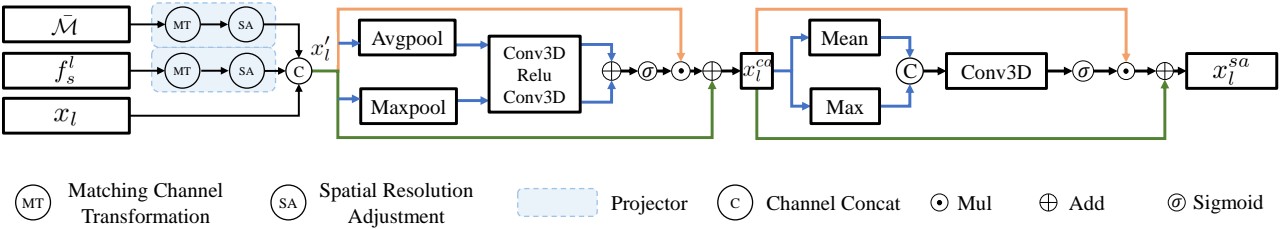

Figure 5. Design of the Residual Channel-Spatial Attention (RCSA) module for dual-stream feature guidance.

reconstructed features, without decoding them back to the image domain. This design is based on the fact that these methods already produce semantically meaningful representations in latent space. In other words, we leverage the multi-layer decoder features directly for cost volume construction, instead of relying on an external pre-trained encoder (e.g., DINO) as done in GLAD+Ours and AnomalDF+Ours.

(v) **Residual Channel-Spatial Attention (RCSA) module**. Fig. 5 illustrates the RCSA module (Eq. 4), which generates two attention tensors: a channel attention tensor of shape $(B, C', 1, 1, 1)$ and a spatial attention tensor of shape $(B, 1, D', H', W')$, where $D'$ represents the depth of features at each layer. These tensors refine feature representations across both matching and spatial dimensions. The RCSA module enhances global feature matching via residual channel attention and improves pixel-level anomaly localization through residual spatial attention. Notably, the residual connections preserve anomaly-relevant features, facilitating progressive denoising and precise anomaly detection.

## B. Challenges in GLAD: Visual Analysis and Comparison with Our Method

As discussed in the main text, the final anomaly score map in Unsupervised Anomaly Detection (UAD) relies on matching the input sample with normal templates. Despite advancements, anomaly regions often retain elevated matching noise. This section identifies key challenges in reconstruction-based methods.

(i) Asymmetry in multi-class reconstruction induces significant matching noise artifacts. As shown in Fig. 6 (a), a unified multi-class model must reconstruct diverse anomalies, frequently distorting shape, texture, or orientation, resulting in asymmetric feature matching and significant matching noise artifacts. This issue is also prevalent in embedding-based methods (Roth et al., 2022; Damm et al., 2025; Zhong et al., 2022). Our approach mitigates it by leveraging input feature guidance within dual-stream attention, guiding the filtering model to focus on spatial structures while preserving edge details. This effectively enhances anomaly localization, making it compatible with both reconstruction-based and embedding-based models.

(ii) The "identical shortcut" issue weakens anomaly residual signals. Reconstruction-based methods suffer from anomaly information leakage, known as the identical shortcut issue, where anomalies persist in reconstructed outputs. As shown in Fig. 6 (b), a hazelnut's small-hole anomaly remains visible after reconstruction, weakening residual anomaly signals and hindering detection. In contrast, the intermediate outputs of the multi-step denoising process primarily reconstruct low-frequency features (Yang et al., 2023) (e.g., normal structures) at earlier stages. As the reconstruction progresses, the identical shortcut issue may gradually reintroduce anomalous information, making anomalies more apparent in later steps. Motivated by this observation, our matching cost volume and filtering network integrate multi-step reconstruction results, significantly enhancing anomaly detection.

(iii) Widespread noise interference in matching between templates and input images. As shown in Fig. 6(c), the matching process inevitably introduces noise, despite anomalies being reconstructed as normal. In contrast, our cost volume filtering network effectively suppresses these noise artifacts, enhancing anomaly region localization.

## C. Progressive Matching Noise Suppression via CostFilter-AD

Beyond the visualized coarse-to-fine noise suppression in Fig. 2 (lower-left), additional results are presented in Fig. 7 and Fig. 8, illustrating spatial anomaly features extracted by RCSA modules across the first three encoder and last three decoder layers. Following channel and spatial attention refinement, the most anomaly-relevant channel—identified by the highest attention weight—is indexed. The selected features are then depth-wise aggregated into a score map of resolution $H' \times W'$,

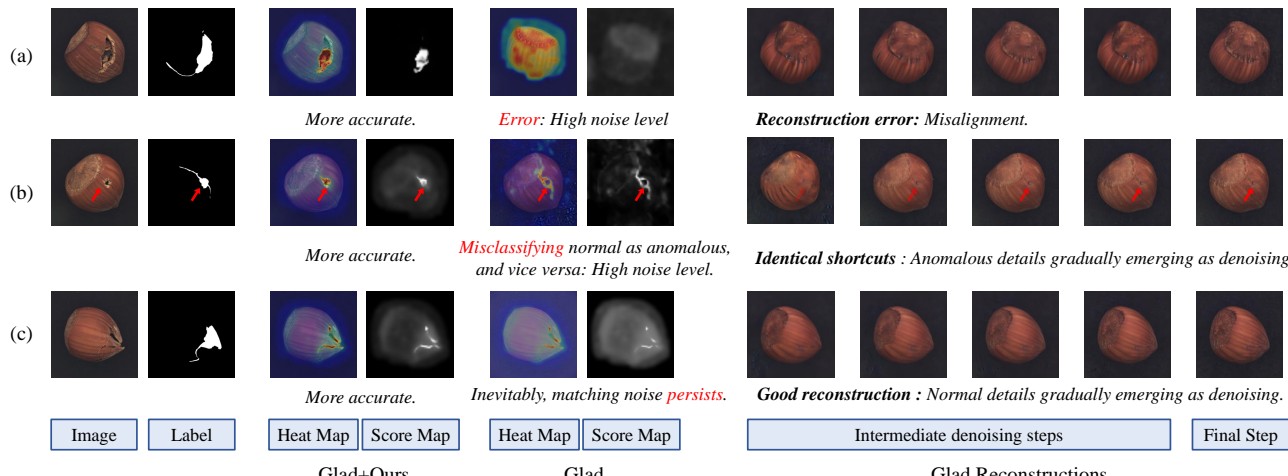

Figure 6. Visualization of challenges in the diffusion-based reconstruction model GLAD (Yao et al., 2024) and advancements enabled by our approach.

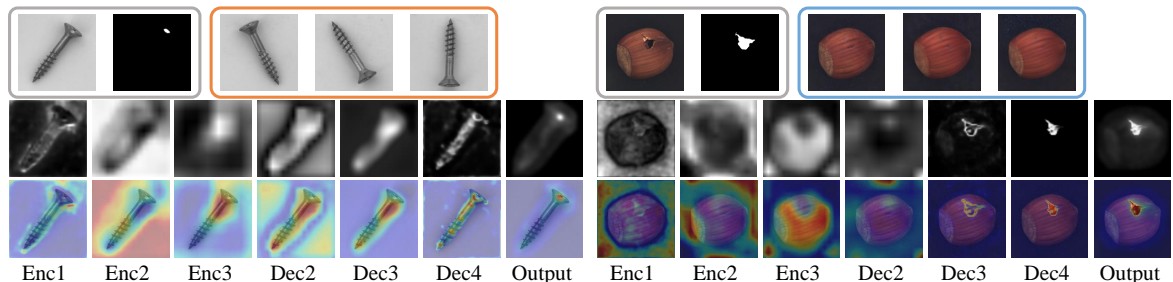

Figure 7. Progressive noise mitigation and coarse-to-fine anomaly localization across filtering network layers. The first row illustrates the anomaly image, ground truth, and three templates from either embedding-based (screw) or reconstruction-based (hazelnut) methods. The subsequent rows depict the corresponding anomaly score maps and localization heatmaps.

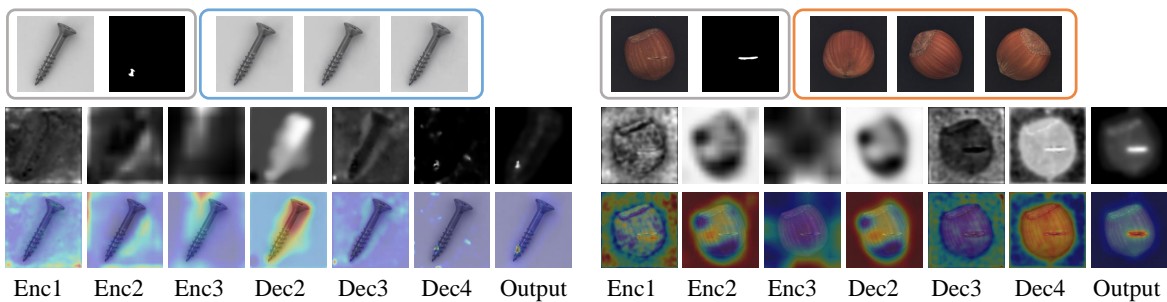

Figure 8. Progressive noise mitigation and coarse-to-fine anomaly localization across filtering network layers. The first row illustrates the anomaly image, ground truth, and three templates from either reconstruction-based (screw) or embedding-based (hazelnut) methods. The subsequent rows depict the corresponding anomaly score maps and localization heatmaps.

*Table 8.* AnomalDF vs. AnomalDF+Ours under a comprehensive setting on MVTec-AD and VisA.

| ID | Dataset | Method | Resize | Temples | I-AUROC | I-AP | I-F1max | P-AUROC | P-AP | P-F1max | P-AUPRO |
|----|---------|--------|--------|---------|---------|------|---------|---------|------|---------|---------|
| 1 | | AnomalDF | 256 | 3 | 96.8 | 98.6 | 97.1 | 98.1 | 61.3 | 60.8 | 93.6 |
| 2 | | +Ours | 256 | 3 | 98.5 | 99.4 | 97.8 | 98.8 | 67.8 | 64.9 | 94.2 |
| 3 | | AnomalDF | 256 | Full | 99.0 | 99.3 | 98.4 | 97.5 | – | 58.7 | 91.7 |
| 4 | MVTec-AD | +Ours | 256 | Full | 99.3 | 99.8 | 98.6 | 98.9 | 68.7 | 65.5 | 96.6 |
| 5 | | AnomalDF | 448 | Full | 99.3 | 99.7 | 98.8 | 97.9 | – | 61.8 | 92.9 |
| 6 | | +Ours | 448 | Full | 99.5 | 99.8 | 98.9 | 99.0 | 72.4 | 68.4 | 95.4 |
| 7 | | AnomalDF | 672 | Full | 99.5 | 99.8 | 99.0 | 98.2 | – | 64.3 | 95.0 |
| 8 | | +Ours | 672 | Full | 99.6 | 99.9 | 99.0 | 99.1 | 74.4 | 69.7 | 96.3 |
| 9 | | AnomalDF | 256 | 3 | 90.5 | 91.4 | 86.2 | 97.4 | 39.6 | 40.4 | 86.3 |
| 10 | | +Ours | 256 | 3 | 94.3 | 95.1 | 90.6 | 99.2 | 44.6 | 45.5 | 84.5 |
| 11 | | AnomalDF | 256 | Full | 94.6 | 95.7 | 90.9 | 98.3 | – | 44.3 | 86.7 |
| 12 | VisA | +Ours | 256 | Full | 95.5 | 96.3 | 91.5 | 99.4 | 45.9 | 46.6 | 87.0 |
| 13 | | AnomalDF | 448 | Full | 97.2 | 97.6 | 93.7 | 98.7 | – | 50.5 | 95.0 |
| 14 | | +Ours | 448 | Full | 97.4 | 97.7 | 93.8 | 99.4 | 42.2 | 53.6 | 95.2 |
| 15 | | AnomalDF | 672 | Full | 97.6 | 97.2 | 94.3 | 98.9 | – | 53.8 | 96.1 |
| 16 | | +Ours | 672 | Full | 97.8 | 98.0 | 94.6 | 99.4 | 47.6 | 54.5 | 96.4 |

which is subsequently upsampled to generate layer-wise anomaly detection heatmaps.

Using the screw as an example, our filtering network iteratively refines anomaly localization, whether based on randomly sampled normal templates with diverse orientations or multi-step denoised images from reconstruction models. This refinement progresses from a global to a local scale while systematically suppressing matching noise at each layer.

## D. Evaluation Under the Original AnomalyDINO Full-shot Settings

In our main paper, AnomalDF and AnomalDF+Ours were trained with 3 randomly sampled reference templates per input and a resolution of $256 \times 256$, and evaluated using a similarly limited number of templates, offering a trade-off between template diversity and memory efficiency. Exp. ID 1, 2, 9, and 10 in Table 8 report the corresponding results.

In contrast, the original full-shot setting of AnomalyDINO (Damm et al., 2025) utilizes the entire training set as reference templates and resizes images to larger resolution. To ensure a fair and thorough comparison, we further conducted evaluations under the original full-shot setting of AnomalyDINO. Exp. ID 3 to 8 and Exp. ID 11 to 16 in Table 8 report results on MVTec-AD and VisA, respectively. Notably, to mitigate storage and compute overhead, we directly reuse the models trained in Exp. ID 2 and 10, and test them under different resolutions and template amounts, without additional retraining. Despite this constraint, CostFilter-AD consistently improved the performance of AnomalyDINO across various resolutions and datasets. Remarkably, our method at lower resolution (e.g., $448 \times 448$) may match or outperform the original AnomalDF baseline at higher resolution (e.g., $672 \times 672$), demonstrating its effectiveness. These results highlight the scalability and generalization capability of our plug-in method, even under varied operational constraints.

## E. Expanded Qualitative Analysis of Our Multi-class UAD Models on MVTec-AD

To provide a more comprehensive validation of our method, we present additional KDE (Kernel Density Estimation) curves (Parzen, 1962) illustrating the distribution of anomaly detection logits across 15 categories in the MVTec-AD dataset, which serves as a supplementary analysis to Fig. 1. As shown in Fig. 9, the green curves represent the logits distribution of our method (along with the boxplot), while the yellow curves correspond to the baselines. In each KDE curve, the left peak represents the normal feature distribution, whereas the right peak corresponds to anomalous features. Greater peak separation and reduced distribution overlap signify enhanced anomaly separability, improving detection performance.

Furthermore, we visualize anomaly detection heatmaps for each category and compare them against the three baseline methods, as shown in Fig. 10. These visualizations provide compelling evidence of the effectiveness of our approach.

## F. Expanded Qualitative Analysis of Our Multi-class UAD Models on VisA

Similarly, we visualize the KDE (Kernel Density Estimation) curves (Parzen, 1962) of anomaly detection logits across 12 anomaly categories in the VisA dataset (Fig. 11). Additionally, we present anomaly detection heatmaps for each category and evaluate them against three competitive baselines (Fig. 12). Results demonstrate that our approach attenuates prevalent matching noise in existing methods, while achieving precise pixel-level localization and accurate image-level anomaly detection, further validating its robustness and generalizability.

## G. Comprehensive Quantitative Analysis of Our Multi-class UAD Models on MVTec-AD

Table 9 to Table 12 presents the image-level detection and pixel-level localization performance of our multi-class model on 15 MVTec-AD anomaly categories. Our method consistently enhances the three latest baselines including UniAD (You et al., 2022), GLAD (Yao et al., 2024), HVQ-Trans (Lu et al., 2023), AnomalDF (Damm et al., 2025), and Dinomaly (Guo et al., 2025), which achieves new state-of-the-art results.

## H. Comprehensive Quantitative Analysis of Our Multi-class UAD Models on VisA

Table 13 to Table 16 summarize image-level detection and pixel-level localization results for our multi-class model on 12 anomaly categories in the VisA dataset. Our method consistently demonstrates superior anomaly detection performance across various evaluated categories, outperforming five baselines and achieving SOTA results across multiple metrics.

## I. Comprehensive Quantitative Analysis of Our Multi-class UAD Models on BTAD

Table 17 and Table 18 report image-level detection and pixel-level localization performance on the BTAD dataset, which includes three distinct anomaly categories. Results are shown for our proposed multi-class UAD model as well as two representative baselines: UniAD (You et al., 2022) and Dinomaly (Guo et al., 2025).

## J. Comprehensive Quantitative Analysis of Our Multi-class UAD Models on MPDD

Table 19 and Table 20 summarize the image- and pixel-level performance of our multi-class UAD model compared to UniAD (You et al., 2022) and Dinomaly (Guo et al., 2025) on the MPDD dataset, which contains six challenging anomaly categories.

*Table 9.* Multi-class anomaly localization results on MVTec-AD using I-AUROC/P-AUROC metrics. The best and second-best results are **bolded** and underlined, respectively.

| Method → | UniAD | UniAD+Ours | GLAD | GLAD+Ours | HVQ-Trans | HVQ-Trans+Ours | AnomalDF | AnomalDF+Ours | Dinomaly | Dinomaly+Ours |
|---|---|---|---|---|---|---|---|---|---|---|
| Category ↓ | NeurIPS'22 | Ours | ECCV'24 | Ours | NeurIPS'23 | Ours | WACV'25 | Ours | CVPR'25 | Ours |
| Bottle | 99.7 / 98.0 | **100.0** / 98.0 | 99.8 / 97.8 | **100.0** / 98.4 | **100.0** / 98.8 | **100.0** / 99.3 | **100.0** / 99.1 | **100.0**/ **100.0** | **100.0** / 99.1 | **100.0** / 99.2 |
| Cable | 95.2 / 97.4 | 99.2 / 97.2 | 98.7 / 93.4 | 98.0 / 96.3 | 99.5 / 98.2 | 99.8 / 98.2 | 99.6 / 98.3 | 99.3 / 98.2 | **100.0** / 98.2 | **100.0** / **98.4** |
| Capsule | 93.4 / 98.7 | 96.3 / 98.7 | 96.5 / 99.1 | 94.3 / **99.2** | 95.4 / 98.6 | 96.4 / 98.9 | 89.7 / 99.1 | 96.1 / **99.2** | 97.9 / 98.7 | **98.2** / 98.8 |
| Hazelnut | **100.0** / 98.1 | **100.0** / 98.5 | 97.0 / 98.9 | 99.4 / 99.1 | 99.9 / 98.8 | **100.0** / 99.2 | 99.9 / **99.6** | **100.0** / 99.5 | **100.0** / 99.4 | **100.0** / 99.5 |
| Metal Nut | 99.5 / 93.7 | 99.6 / 94.6 | 99.9 / 97.3 | **100.0** / **99.2** | **100.0** / 96.2 | **100.0** / 97.9 | **100.0** / 96.7 | **100.0** / 99.0 | **100.0** / 97.0 | **100.0** / 97.4 |
| Pill | 94.8 / 96.2 | 96.8 / 97.1 | 94.5 / 97.9 | 97.9 / 97.8 | 96.7 / 97.2 | 96.9 / 96.5 | 97.2 / 96.7 | 98.9 / **98.4** | **99.2** / 97.8 | **99.2** / 97.8 |
| Screw | 91.7 / 98.8 | 95.1 / 98.7 | 93.4 / 99.6 | 95.4 / 99.6 | 95.8 / 98.4 | 95.3 / 99.0 | 74.3 / 97.6 | 88.5 / 99.0 | 98.4 / 99.6 | **98.6** / **99.7** |
| Toothbrush | 92.8 / 98.4 | 98.9 / 98.9 | 99.7 / **99.2** | 99.7 / 99.1 | 91.1 / 98.5 | **100.0** / 99.0 | 99.7 / **99.2** | 99.7 / **99.2** | **100.0** / 98.9 | **100.0** / 98.9 |
| Transistor | 99.5 / 98.0 | 99.8 / 98.0 | 99.4 / 90.9 | 99.5 / 91.6 | **100.0** / **98.3** | **100.0** / 97.1 | 96.5 / 95.8 | 97.8 / 97.5 | 99.1 / 93.2 | 99.1 / 93.3 |
| Zipper | 98.2 / 97.7 | 99.9 / 97.7 | 96.4 / 93.0 | 99.2 / 97.7 | 97.9 / 97.5 | 98.9 / 98.3 | 98.8 / 94.3 | 98.9 / 96.7 | **100.0** / 99.2 | **100.0** / **99.3** |
| Carpet | 99.8 / 98.4 | 99.9 / 98.4 | 97.2 / 98.9 | **100.0** / 99.1 | 99.9 / 98.7 | **100.0** / 98.5 | 99.9 / 99.4 | 99.9 / **99.6** | 99.8 / 99.3 | 99.9 / 99.4 |
| Grid | 98.7 / 97.3 | 99.9 / 98.7 | 95.1 / 98.2 | **100.0** / **99.5** | 96.0 / 97.1 | 99.4 / 98.3 | 98.7 / 97.8 | **100.0** / 99.5 | 99.7 / 99.4 | 99.8 / **99.5** |
| Leather | **100.0** / 98.7 | **100.0** / 99.4 | 99.5 / 99.2 | **100.0** / 99.6 | **100.0** / 98.8 | **100.0** / 99.3 | **100.0** / **99.7** | **100.0** / **99.7** | **100.0** / 99.3 | **100.0** / 99.4 |
| Tile | 99.5 / 91.8 | **100.0** / 95.3 | **100.0** / 97.8 | **100.0** / 99.4 | 99.1 / 92.8 | **100.0** / 95.0 | **100.0** / 98.5 | **100.0** / **99.6** | **100.0** / 98.1 | **100.0** / 98.3 |
| Wood | 98.5 / 93.1 | 98.9 / 94.0 | 95.4 / 97.3 | 97.4 / 98.2 | 97.6 / 97.4 | 98.5 / 97.9 | 98.0 / 98.1 | 99.0 / **98.8** | **99.9** / 97.6 | **99.9** / 97.7 |
| Mean | 97.5 / 96.9 | 99.0 / 97.5 | 97.5 / 97.3 | 98.7 / 98.2 | 97.9 / 97.4 | 99.0 / 97.9 | 96.8 / 98.1 | 98.5 / **98.8** | 99.6 / 98.3 | **99.7**/ 98.4 |

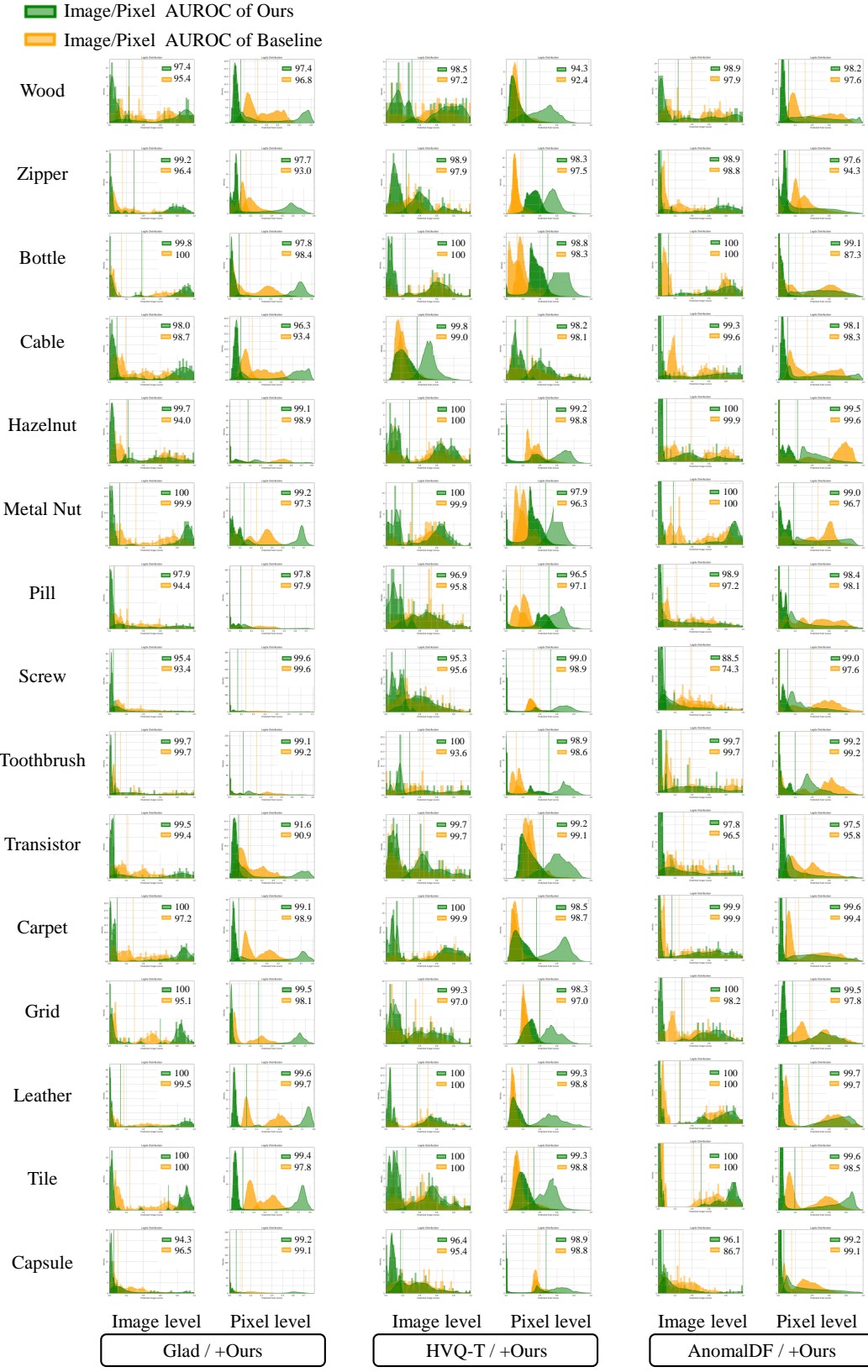

*Figure 9.* Quantitative comparison on MVTec-AD using KDE curves of image- and pixel-level anomaly logits. Each two-column pair (from left to right) compares GLAD, HVQ-Trans, and AnomalDF with our method, where the first and second columns show image- and pixel-level APROC, respectively.

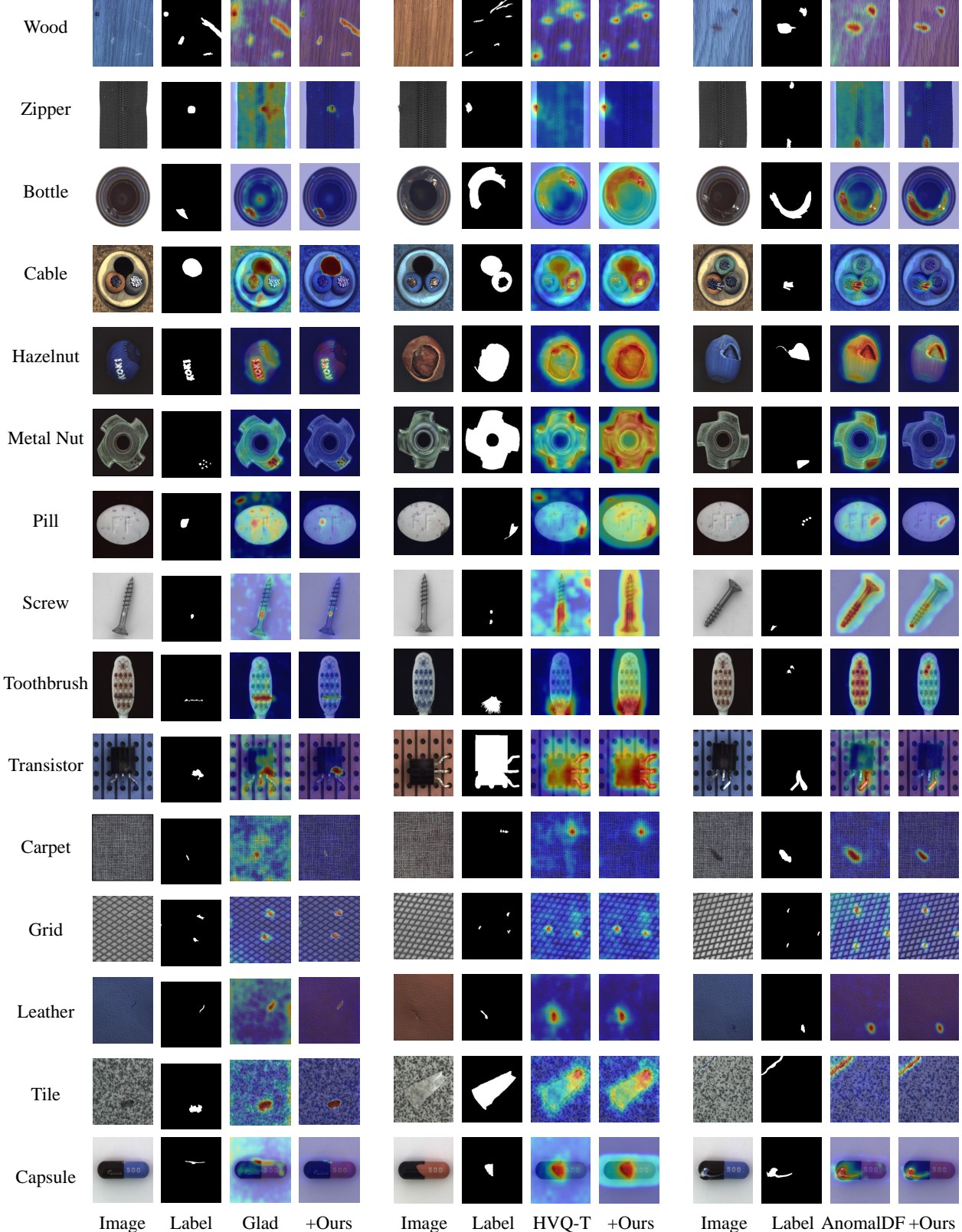

*Figure 10.* Quantitative examples of anomaly localization on MVTec-AD. The comparison is divided into three groups, each following the same left-to-right order: input anomaly, ground truth mask, anomaly map predicted by GLAD, HVQ-Trans, or AnomalDF, and the anomaly map obtained with our method integrated.

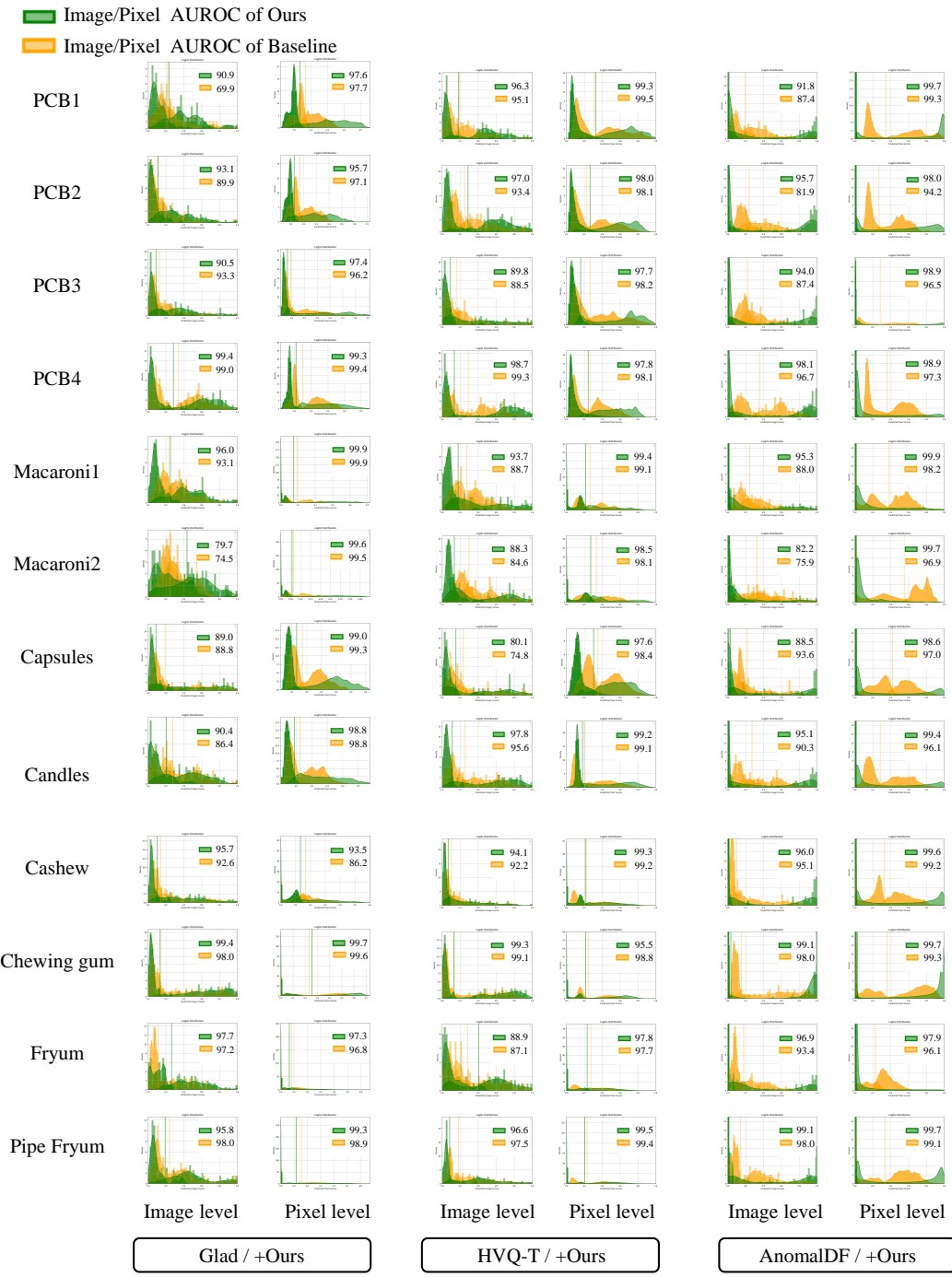

*Figure 11.* Quantitative comparison on VisA using KDE curves of image- and pixel-level anomaly logits. Each two-column pair (from left to right) compares GLAD, HVQ-Trans, and AnomalDF with our method, where the first and second columns show image- and pixel-level APROC, respectively.

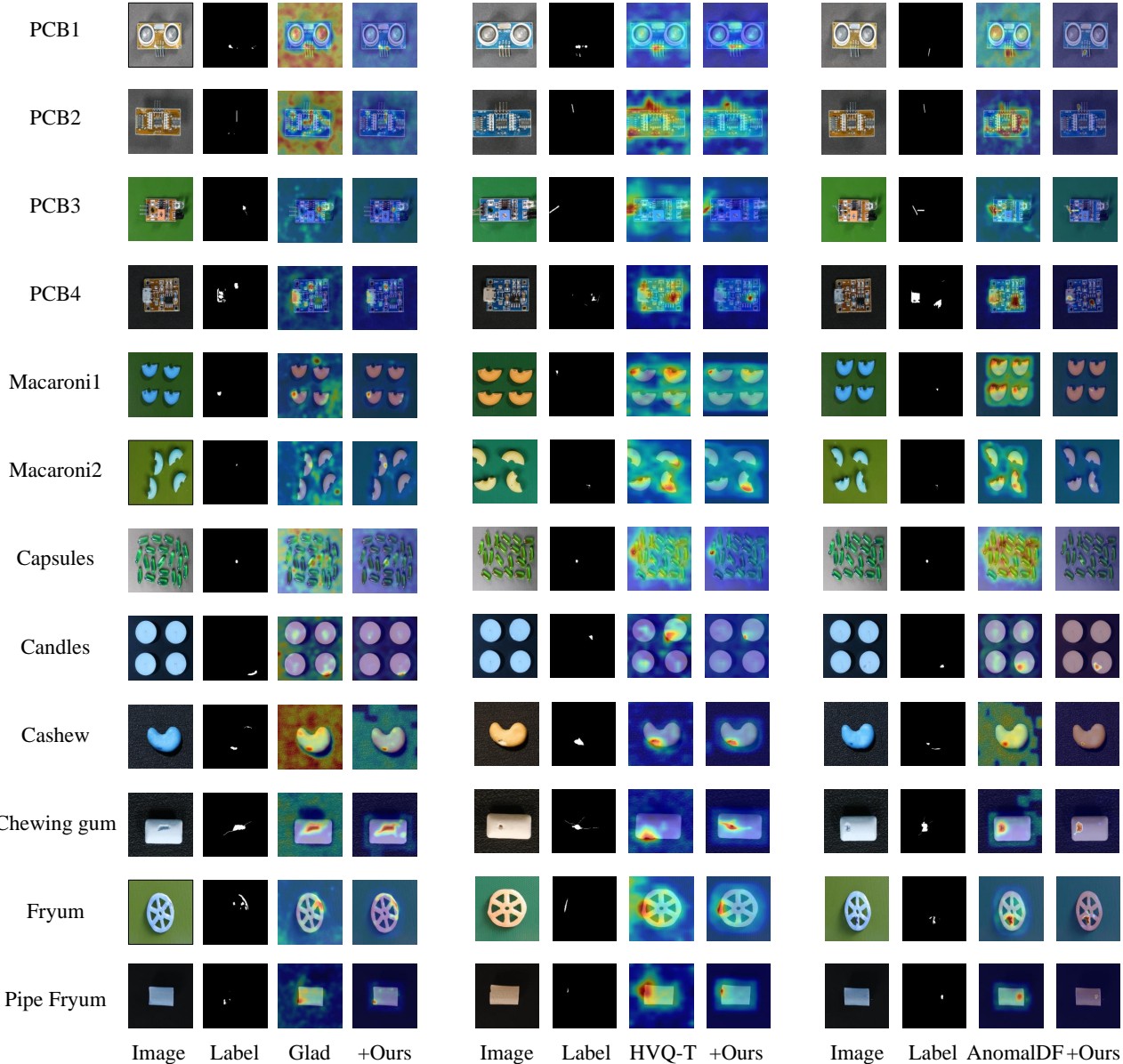

*Figure 12.* Quantitative examples of anomaly localization on VisA. The comparison is divided into three groups, each following the same left-to-right order: input anomaly, ground truth mask, anomaly map predicted by GLAD, HVQ-Trans, or AnomalDF, and the anomaly map obtained with our method integrated.

*Table 10.* Multi-class anomaly detection and localization results on MVTec-AD using I-AP/P-AP metrics.

| Method → | | UniAD | UniAD+Ours | GLAD | GLAD+Ours | HVQ-Trans | HVQ-Trans+Ours | AnomalDF | AnomalDF+Ours | Dinomaly | Dinomaly+Ours |
|---|---|---|---|---|---|---|---|---|---|---|---|
| Category ↓ | | NeurIPS'22 | Ours | ECCV'24 | Ours | NeurIPS'23 | Ours | WACV'25 | Ours | CVPR'25 | Ours |
| Objects | Bottle | **100.0** / 66.4 | **100.0** / 81.2 | **100.0** / 80.9 | **100.0** / 79.2 | **100.0** / 73.9 | **100.0** / 81.4 | **100.0** / 87.3 | 100.0/ 85.9 | **100.0** / 88.3 | **100.0** / 88.2 |
| | Cable | 97.3 / 47.6 | 99.5 / 57.2 | 99.3 / 51.2 | 98.8 / 64.1 | 99.7 / 54.2 | 99.8 / 58.0 | 99.8 / 69.3 | 99.6 / **72.3** | **100.0** / 66.7 | **100.0** / 67.3 |
| | Capsule | 98.4 / 44.5 | 99.2 / 51.4 | 99.2 / 49.1 | 98.8 / 53.1 | 99.0 / 44.0 | 99.2 / 49.2 | 97.3 / 45.9 | 99.1 / 45.4 | 99.6 / 60.7 | **99.7** / 60.7 |
| | Hazelnut | **100.0** / 54.6 | **100.0** / 70.4 | 98.2 / 68.0 | 99.6 / 75.8 | **100.0** / 63.1 | **100.0** / 72.4 | 99.9 / 79.0 | **100.0** / 77.6 | **100.0** / 81.9 | **100.0** / 81.9 |
| | Metal Nut | 99.9 / 50.7 | 99.9 / 69.1 | **100.0** / 81.8 | **100.0** / **93.1** | **100.0** / 65.0 | **100.0** / 79.0 | **100.0** / 78.6 | **100.0** / 92.0 | **100.0** / 80.1 | **100.0** / 80.2 |
| | Pill | 99.0 / 44.3 | 99.4 / 58.0 | 99.0 / 73.9 | 99.6 / 69.6 | 99.4 / 57.3 | 99.4 / 59.6 | 99.5 / **78.6** | 99.8 / 76.2 | 99.2 / 75.9 | 99.2 / 75.8 |
| | Screw | 97.1 / 29.4 | 98.2 / 32.1 | 98.0 / 47.8 | 98.6 / 40.8 | 98.3 / 28.6 | 98.3 / 33.5 | 88.0 / 12.5 | 96.3 / 31.4 | **99.9** / 75.9 | **99.9** / 75.8 |
| | Toothbrush | 96.9 / 38.3 | 99.6 / **67.6** | 99.9 / 45.0 | 99.9 / 44.3 | 96.1 / 40.0 | **100.0** / 51.6 | 99.9 / 46.9 | 99.9 / 44.0 | **100.0** / 52.7 | **100.0** / 52.8 |
| | Transistor | 99.3 / 65.2 | 99.7 / 61.9 | 99.2 / 58.9 | 99.3 / 62.5 | **100.0** / 74.5 | **100.0** / **76.6** | 96.1 / 62.4 | 97.4 / 73.2 | 98.4 / 59.6 | 98.5 / 59.6 |
| | Zipper | 99.5 / 40.0 | **100.0** / 64.7 | 98.9 / 40.9 | 99.8 / 66.5 | 99.4 / 39.7 | 99.7 / 55.1 | 99.7 / 44.0 | 99.7 / 55.0 | **100.0** / 79.2 | **100.0** / 79.2 |
| Textures | Carpet | 99.9 / 50.7 | **100.0** / 60.5 | 99.1 / 72.2 | **100.0** / 78.6 | **100.0** / 57.6 | **100.0** / 64.7 | **100.0** / 76.2 | **100.0** / 81.6 | **100.0** / 68.5 | **100.0** / 68.6 |
| | Grid | 99.6 / 22.8 | **100.0** / 39.8 | 93.6 / 10.2 | **100.0** / 43.8 | 98.7 / 25.0 | 99.8 / 34.0 | 99.3 / 31.0 | **100.0** / 42.8 | 99.9 / 54.5 | **100.0** / 54.5 |
| | Leather | **100.0** / 32.4 | **100.0** / 66.1 | 99.8 / 61.7 | **100.0** / 62.5 | **100.0** / 34.5 | **100.0** / 47.4 | **100.0** / 60.2 | **100.0** / 61.1 | **100.0** / 51.9 | **100.0** / 52.4 |
| | Tile | 99.8 / 42.1 | **100.0** / 68.0 | **100.0** / 70.3 | **100.0** / 92.2 | 99.6 / 43.6 | **100.0** / 56.0 | **100.0** / 76.4 | **100.0** / 96.0 | **100.0** / 78.6 | **100.0** / 79.3 |
| | Wood | 99.6 / 37.0 | 99.7 / 57.6 | 98.5 / 70.6 | 99.2 / 77.1 | 99.2 / 39.9 | 99.5 / 52.4 | 99.3 / 72.7 | 99.7 / **82.1** | **100.0** / 73.0 | **100.0** / 73.2 |
| | Mean | 99.1 / 44.5 | 99.7 / 60.5 | 98.8 / 58.8 | 99.6 / 66.8 | 99.3 / 49.4 | 99.7 / 58.1 | 98.6 / 61.3 | 99.4 / 67.8 | **99.8** / 68.7 | **99.8** / **68.9** |

*Table 11.* Multi-class anomaly detection and localization results on MVTec-AD using I-F1max/P-F1max metrics.

| Method → | | UniAD | UniAD+Ours | GLAD | GLAD+Ours | HVQ-Trans | HVQ-Trans+Ours | AnomalDF | AnomalDF+Ours | Dinomaly | Dinomaly+Ours |
|---|---|---|---|---|---|---|---|---|---|---|---|
| Category ↓ | | NeurIPS'22 | Ours | ECCV'24 | Ours | NeurIPS'23 | Ours | WACV'25 | Ours | CVPR'25 | Ours |
| Objects | Bottle | **100.0** / 68.9 | **100.0** / 73.7 | **100.0** / 75.5 | 99.2 / 72.6 | **100.0** / 71.6 | **100.0** / 77.6 | **100.0** / 80.2 | **100.0**/ 77.0 | **100.0** / 83.8 | **100.0** / 83.6 |
| | Cable | 89.9 / 55.4 | 97.3 / 59.8 | 97.3 / 53.4 | 95.2 / 60.5 | 97.8 / 60.8 | 98.4 / 63.0 | 97.8 / 67.0 | 97.9 / 66.2 | 99.5 / 69.1 | **99.6** / **69.8** |
| | Capsule | 95.5 / 48.2 | 95.6 / 53.2 | 96.8 / 51.2 | 95.5 / 51.6 | 96.8 / 48.3 | 96.4 / 53.0 | 94.3 / 48.9 | 96.4 / 50.3 | **97.3** / 60.5 | **97.3** / 60.5 |
| | Hazelnut | **100.0** / 55.9 | **100.0** / 67.8 | 94.4 / 63.8 | 97.2 / 69.4 | 99.3 / 63.1 | 99.3 / 70.8 | **100.0** / 75.5 | **100.0** / 69.9 | **100.0** / 76.8 | **100.0** / 76.8 |
| | Metal Nut | 98.9 / 66.3 | 98.9 / 66.6 | 99.5 / 82.4 | 100 / **87.0** | 99.5 / 74.3 | 99.5 / 82.1 | **100.0** / 79.5 | **100.0** / 69.8 | **100.0** / 86.9 | **100.0** / 86.9 |
| | Pill | 95.6 / 53.7 | 96.8 / 57.1 | 94.6 / 69.9 | 98.6 / 69.0 | 95.9 / 62.1 | 96.9 / 61.2 | 97.1 / **79.5** | 99.8 / 69.8 | 98.3 / 71.4 | 98.3 / 71.3 |
| | Screw | 91.8 / 38.0 | 93.9 / 36.0 | 92.2 / 47.6 | 93.9 / 38.8 | 94.6 / 36.8 | 94.5 / 40.5 | 87.2 / 19.4 | 88.4 / 36.0 | 95.9 / 59.6 | **96.1** / 59.6 |
| | Toothbrush | 95.2 / 49.7 | 96.7 / 68.6 | 98.4 / 57.4 | 98.4 / 55.3 | 95.2 / 50.9 | **100.0** / 61.9 | 98.4 / 57.7 | 98.4 / 57.3 | **100.0** / 63.0 | **100.0** / 63.1 |
| | Transistor | 97.5 / 67.1 | 98.8 / 58.5 | 95.0 / 58.3 | 95.2 / 59.6 | 98.8 / 72.1 | **100.0** / **74.1** | 89.7 / 59.4 | 91.6 / 68.0 | 96.3 / 57.9 | 96.6 / 57.8 |
| | Zipper | 97.1 / 49.7 | 99.2 / 63.1 | 95.6 / 46.2 | 97.5 / 62.2 | 97.1 / 48.9 | 98.3 / 59.5 | 97.9 / 49.3 | 97.9 / 54.2 | **100.0** / 75.4 | **100.0** / **75.6** |
| Textures | Carpet | 99.4 / 51.1 | 98.9 / 60.8 | 96.6 / 67.9 | **100.0** / 72.5 | 99.4 / 58.1 | **100.0** / 63.3 | 99.4 / 67.7 | 99.4 / 71.4 | 98.9 / 71.2 | 98.9 / 71.2 |
| | Grid | 98.2 / 28.4 | 99.1 / 47.1 | 98.3 / 24.1 | **100.0** / 49.4 | **100.0** / 31.1 | 98.2 / 40.6 | 96.6 / 37.4 | **100.0** / 47.3 | 99.1 / 57.4 | 99.5 / **57.6** |
| | Leather | **100.0** / 34.1 | **100.0** / **62.2** | 98.4 / 60.7 | 98.6 / 60.6 | **100.0** / 37.0 | **100.0** / 50.0 | **100.0** / 57.4 | **100.0** / 59.3 | **100.0** / 53.6 | **100.0** / 53.9 |
| | Tile | 99.8 / 50.2 | **100.0** / 67.0 | **100.0** / 71.5 | **100.0** / 88.2 | 96.5 / 37.0 | **100.0** / 63.5 | **100.0** / 76.6 | **100.0** / 88.6 | **100.0** / 76.0 | **100.0** / 76.2 |
| | Wood | **99.6** / 41.2 | 96.7 / 57.7 | 95.1 / 65.2 | 95.9 / 70.3 | 95.9 / 45.6 | 97.5 / 56.5 | 98.4 / 65.4 | 98.3 / **73.1** | 99.2 / 68.7 | 99.2 / 68.7 |
| | Mean | 97.0 / 50.5 | 98.1 / 59.9 | 96.8 / 59.7 | 97.8 / 64.4 | 97.4 / 54.3 | 98.6 / 61.2 | 97.1 / 60.8 | 97.8 / 64.9 | 99.0 / 68.7 | **99.1** / **68.9** |

*Table 12.* Multi-class anomaly localization results on MVTec-AD using P-AURPO metrics.

| | Method → | UniAD | UniAD+Ours | GLAD | GLAD+Ours | HVQ-Trans | HVQ-Trans+Ours | AnomalDF | AnomalDF+Ours | Dinomaly | Dinomaly+Ours |
|---|---|---|---|---|---|---|---|---|---|---|---|
| | Category ↓ | NeurIPS'22 | Ours | ECCV'24 | Ours | NeurIPS'23 | Ours | WACV'25 | Ours | CVPR'25 | Ours |
| Objects | Bottle | 93.2 | 94.3 | 96.1 | 90.8 | 94.4 | 96.1 | 97.5 | 95.3 | **96.6** | 96.5 |
| | Cable | 86.0 | 76.8 | 89.6 | 89.8 | 89.6 | 91.7 | **94.2** | 92.5 | 93.7 | **94.2** |
| | Capsule | 91.1 | 60.0 | 96.1 | 93.1 | 89.9 | 92.9 | 92.5 | 92.8 | **97.3** | 97.1 |
| | Hazelnut | 92.8 | 87.2 | 90.8 | 93.0 | 93.8 | 92.9 | 92.5 | 92.8 | **96.9** | 96.9 |
| | Metal Nut | 82.4 | 87.2 | 94.2 | 96.3 | 90.6 | 93.3 | 94.7 | 96.2 | **97.5** | 97.5 |
| | Pill | 95.3 | 71.0 | 94.3 | 96.8 | 94.9 | 96.1 | 94.7 | 95.8 | **97.5** | 97.5 |
| | Screw | 94.9 | 60.0 | 96.7 | 96.4 | 92.3 | 95.4 | 89.4 | 92.9 | **98.3** | 98.3 |
| | Toothbrush | 87.7 | 80.5 | 95.6 | 96.0 | 87.4 | 89.6 | **96.1** | 96.0 | 95.0 | 95.3 |
| | Transistor | 94.3 | 87.4 | 86.5 | 85.8 | **94.8** | 93.7 | 84.2 | 86.0 | 75.9 | 76.5 |
| | Zipper | 86.7 | 92.7 | 84.5 | 93.8 | 91.8 | 93.3 | 86.2 | 89.4 | 97.0 | **97.2** |
| Textures | Carpet | 94.4 | 89.0 | 95.3 | 97.1 | 94.8 | 95.4 | 97.6 | 98.1 | 97.5 | **97.7** |
| | Grid | 91.9 | 95.3 | 92.7 | **97.5** | 90.3 | 93.4 | 90.0 | 96.9 | 96.9 | 97.1 |
| | Leather | 97.1 | 97.9 | 97.0 | 96.9 | 97.7 | **98.8** | 98.5 | 97.5 | 97.3 | 97.5 |
| | Tile | 79.0 | 86.6 | 96.8 | 97.8 | 82.6 | 85.5 | 96.7 | 96.5 | **90.9** | 90.7 |
| | Wood | 85.7 | 86.0 | 86.3 | 90.8 | 87.1 | 90.0 | 93.4 | 93.5 | **93.8** | 93.8 |
| | Mean | 90.6 | 91.3 | 92.8 | 94.1 | 91.5 | 93.2 | 93.6 | 94.1 | 94.6 | **94.8** |

*Table 13.* Multi-class anomaly detection and localization results on VisA using I-AUROC/P-AUROC metrics.

| | Method → | UniAD | UniAD+Ours | GLAD | GLAD+Ours | HVQ-Trans | HVQ-Trans+Ours | AnomalDF | AnomalDF+Ours | Dinomaly | Dinomaly+Ours |
|---|---|---|---|---|---|---|---|---|---|---|---|
| | Category ↓ | NeurIPS'22 | Ours | ECCV'24 | Ours | NeurIPS'23 | Ours | WACV'25 | Ours | CVPR'25 | Ours |
| Complex Structure | PCB1 | 95.1/98.9 | 95.4/99.4 | 69.9/97.6 | 90.9/97.7 | 95.2/99.5 | 96.3/99.3 | 87.4/99.3 | 91.8/**99.7** | 99.0/99.5 | **99.1**/99.5 |
| | PCB2 | 96.7/93.0 | 82.8/93.3 | 89.9/97.1 | 93.2/95.7 | 93.3/**98.1** | 97.0/98.0 | 81.9/94.2 | 95.7/98.0 | 99.2/98.0 | **99.4**/98.1 |
| | PCB3 | 88.4/96.5 | 89.6/98.3 | 93.3/96.2 | 90.5/97.4 | 89.0/98.2 | 89.8/97.7 | 87.4/96.5 | 94.0/**98.9** | 98.8/98.4 | 98.8/98.5 |
| | PCB4 | 98.7/98.1 | 99.3/97.8 | 99.0/**99.4** | 99.4/99.3 | 99.3/98.1 | 98.7/97.8 | 96.7/97.3 | 98.1/98.9 | 99.7/98.7 | **99.9**/98.7 |
| Multiple Instances | Macaroni1 | 95.9/99.6 | 92.9/99.3 | 93.1/**99.9** | 96.0/99.9 | 89.4/99.1 | 93.7/99.4 | 88.0/98.2 | 95.3/**99.9** | 97.8/99.6 | 98.0/99.7 |
| | Macaroni2 | 79.1/97.5 | 84.1/98.1 | 74.5/99.5 | 79.7/99.6 | 84.6/98.1 | 88.3/98.5 | 75.9/96.9 | 82.2/99.7 | **95.7**/99.7 | 95.6/99.8 |
| | Capsules | 76.9/95.9 | 75.6/98.2 | 88.8/99.3 | 89.1/99.0 | 76.0/98.4 | 80.1/97.6 | 93.6/97.0 | 88.5/98.7 | 98.6/99.6 | 98.7/99.6 |
| | Candles | 96.2/**99.4** | 96.5/99.1 | 86.4/98.8 | 90.5/98.8 | 95.4/99.1 | 97.8/99.2 | 90.3/96.1 | 95.1/99.4 | 98.8/99.4 | 98.9/99.4 |
| Single Instance | Cashew | 89.1/97.4 | 92.9/98.5 | 92.6/86.2 | 95.7/93.5 | 92.3/98.7 | 94.1/99.3 | 95.1/99.2 | 96.0/99.6 | 98.5/96.7 | 98.4/96.8 |
| | Chewing gum | 96.6/99.3 | 99.0/99.1 | 98.0/99.6 | 99.4/99.7 | 98.8/98.1 | 99.3/99.5 | 98.0/99.3 | 99.1/99.6 | 99.7/99.1 | 99.7/99.2 |
| | Fryum | 91.9/**98.2** | 89.3/97.6 | 97.2/96.8 | 97.7/97.3 | 87.6/97.7 | 94.9/97.7 | 93.4/96.1 | 96.9/98.0 | 99.0/96.6 | 99.0/96.6 |
| | Pipe Fryum | 96.9/98.9 | 97.4/99.1 | 98.1/98.9 | 95.8/99.3 | 97.1/99.4 | 96.6/99.5 | 98.0/99.1 | 99.1/**99.7** | 99.2/99.2 | 99.3/99.2 |
| | Mean | 91.5/98.0 | 92.1/98.6 | 90.1/97.4 | 93.2/98.1 | 91.5/98.5 | 93.4/98.6 | 90.5/97.4 | 94.3/**99.2** | 98.7/98.7 | 98.7/98.8 |

*Table 14.* Multi-class anomaly detection and localization results on VisA using I-AP/P-AP metrics.

| | Method → | UniAD | UniAD+Ours | GLAD | GLAD+Ours | HVQ-Trans | HVQ-Trans+Ours | AnomalDF | AnomalDF+Ours | Dinomaly | Dinomaly+Ours |
|---|---|---|---|---|---|---|---|---|---|---|---|
| | Category ↓ | NeurIPS'22 | Ours | ECCV'24 | Ours | NeurIPS'23 | Ours | WACV'25 | Ours | CVPR'25 | Ours |
| Complex Structure | PCB1 | 93.8/51.5 | 94.4/63.3 | 72.5/38.0 | 88.7/64.5 | 94.5/71.6 | 95.4/71.6 | 87.4/81.3 | 90.6/75.5 | 98.9/87.8 | 98.9/**88.2** |
| | PCB2 | 93.2/10.6 | 93.9/9.3 | 88.9/6.4 | 92.0/6.5 | 94.2/9.5 | 97.1/12.3 | 81.1/12.0 | 96.2/13.1 | 99.2/45.6 | **99.4/46.3** |
| | PCB3 | 89.5/23.8 | 90.2/18.7 | 94.0/25.0 | 90.7/22.4 | 89.4/18.0 | 90.4/25.5 | 90.2/23.3 | 94.5/30.7 | 98.8/41.0 | **99.0/41.4** |
| | PCB4 | 98.6/35.2 | 99.2/33.2 | 98.2/52.6 | 99.4/47.5 | 99.2/31.9 | 98.4/38.2 | 96.3/37.4 | 97.7/32.8 | 99.7/50.1 | **100.0/50.8** |
| Multiple Instances | Macaroni1 | 96.2/16.4 | 93.0/8.6 | 93.1/11.0 | 96.8/16.6 | 89.1/9.7 | 94.1/11.5 | 88.9/10.6 | 95.8/15.7 | 97.2/30.2 | **97.6/30.9** |
| | Macaroni2 | 80.0/4.6 | 84.7/3.7 | 73.8/7.0 | 81.4/6.1 | 83.3/4.0 | 89.3/6.5 | 76.2/5.5 | 81.8/4.6 | **95.5**/24.5 | **95.5/24.8** |
| | Capsules | 87.4/24.1 | 86.4/46.5 | 94.1/47.8 | 94.2/45.3 | 54.0/49.8 | 90.0/45.9 | 96.4/43.3 | 93.0/30.0 | 99.0/66.1 | **99.2**/66.0 |
| | Candles | 96.4/37.2 | 97.0/21.3 | 88.2/29.3 | 91.6/29.7 | 28.6/18.6 | 98.0/29.9 | 90.2/28.1 | 95.9/36.9 | 98.8/43.0 | **99.0/43.8** |
| Single Instance | Cashew | 94.6/27.2 | 96.3/44.7 | 96.4/29.2 | 97.9/57.4 | 96.4/58.6 | 97.5/**64.6** | 97.6/60.2 | 97.8/88.0 | **99.4**/62.5 | **99.4**/63.0 |
| | Chewing gum | 98.4/64.0 | 99.5/59.1 | 99.1/**73.9** | 99.7/83.2 | 99.5/40.9 | 99.6/70.2 | 99.2/65.8 | 99.6/67.2 | 99.9/63.5 | 99.9/66.7 |
| | Fryum | 96.1/51.6 | 94.9/45.6 | 98.9/36.1 | 99.0/42.2 | 94.3/51.0 | 88.9/49.5 | 97.4/46.7 | 98.7/**52.9** | 95.5/52.0 | 99.8/51.9 |
| | Pipe Fryum | 98.6/45.6 | 98.7/54.2 | 99.3/50.1 | 97.9/66.8 | 98.6/61.9 | 98.3/71.5 | 99.2/61.0 | 99.5/80.9 | 99.6/63.8 | 99.8/64.1 |
| | Mean | 93.6/32.7 | 94.0/34.9 | 91.4/33.9 | 94.1/40.7 | 93.4/35.5 | 95.2/41.4 | 91.4/39.6 | 95.1/44.6 | 98.9/52.5 | **99.0/53.2** |

*Table 15.* Multi-class anomaly detection and localization results on VisA using I-F1max/P-F1max metrics.

| Method → Category ↓ | | UniAD NeurIPS'22 | UniAD+Ours Ours | GLAD ECCV'24 | GLAD+Ours Ours | HVQ-Trans NeurIPS'23 | HVQ-Trans+Ours Ours | AnomalDF WACV'25 | AnomalDF+Ours Ours | Dinomaly CVPR'25 | Dinomaly+Ours Ours |
|---|---|---|---|---|---|---|---|---|---|---|---|
| Complex Structure | PCB1 | 91.4 / 55.9 | 93.2 / 62.3 | 70.1 / 44.4 | 86.1 / 60.4 | 91.6 / 63.7 | 91.6 / 66.9 | 82.2 / 68.0 | 86.9 / 75.5 | 96.6 / 80.2 | **96.8 / 80.5** |
| | PCB2 | 86.1 / 20.4 | 88.2 / 16.6 | 83.3 / 14.4 | 87.9 / 14.1 | 88.8 / 16.5 | 92.0 / 21.1 | 76.2 / 23.7 | 90.1 / 25.7 | 97.0 / 48.8 | **97.2 / 48.9** |
| | PCB3 | 81.5 / 29.1 | 83.1 / 25.0 | 87.6 / 27.7 | 84.5 / 27.0 | 81.1 / 22.7 | 82.1 / 28.6 | 80.2 / 38.8 | 88.7 / 34.9 | 95.6 / 45.6 | 95.8 / 45.8 |
| | PCB4 | **98.5** / 38.7 | 97.6 / 35.6 | 98.0 / 52.0 | 97.0 / 50.1 | 97.0 / 36.0 | 97.1 / 39.4 | 91.0 / 30.8 | 95.1 / 35.2 | 98.0 / 52.8 | 98.2 / **53.4** |
| Multiple Instances | Macaroni1 | 90.5 / 25.0 | 87.6 / 17.5 | 85.4 / 19.2 | 89.8 / 26.3 | 86.1 / 19.9 | 86.7 / 19.7 | 79.5 / 17.3 | 88.8 / 23.3 | 94.5 / 38.9 | **95.2 / 39.8** |
| | Macaroni2 | 75.4 / 11.0 | 79.0 / 9.3 | 71.8 / 19.3 | 74.1 / 14.5 | 79.1 / 10.5 | 81.5 / 15.5 | 73.0 / 11.1 | 77.9 / 11.3 | 90.4 / 36.2 | **90.6 / 36.6** |
| | Capsules | 78.0 / 30.6 | 77.1 / 50.5 | 85.9 / 53.3 | 87.3 / 54.7 | 78.0 / 54.0 | 79.4 / 51.4 | 89.8 / 45.6 | 85.1 / 36.1 | 97.1 / 66.8 | **97.7 / 66.9** |
| | Candles | 94.0 / 44.1 | 89.1 / 32.4 | 79.8 / 36.6 | 83.3 / 35.3 | 88.6 / 28.6 | 92.5 / 41.0 | 82.9 / 30.6 | 90.0 / 39.2 | 95.5 / 48.5 | **95.7 / 49.0** |
| Single Instance | Cashew | 87.3 / 35.6 | 91.9 / 49.7 | 90.5 / 38.2 | 92.8 / 58.7 | 91.1 / 61.0 | 91.5 / 66.1 | 92.0 / 60.3 | 94.2 / **81.0** | **96.5** / 60.9 | 95.8 / 61.4 |
| | Chewing gum | 95.5 / 61.2 | 97.5 / 57.2 | 95.5 / 69.6 | 98.0 / 75.9 | 97.0 / 41.6 | 96.9 / 64.7 | 97.5 / 56.9 | 97.5 / 59.7 | 98.0 / 67.4 | **98.1 / 68.6** |
| | Fryum | 89.5 / 55.3 | 87.4 / 54.2 | 95.8 / 43.5 | 95.0 / 44.8 | 85.8 / **56.0** | 86.8 / 54.0 | 92.7 / 45.2 | 95.4 / 54.2 | 96.6 / 53.7 | **97.2** / 53.5 |
| | Pipe Fryum | 93.9 / 53.9 | 95.5 / 58.3 | 97.0 / 55.1 | 94.5 / 63.1 | 93.5 / 64.6 | 93.5 / **71.3** | 97.5 / 56.2 | **98.0** / 70.2 | 97.0 / 65.1 | 97.1 / 65.5 |
| | Mean | 88.5 / 38.4 | 88.9 / 39.1 | 86.7 / 39.4 | 89.2 / 43.7 | 88.1 / 39.6 | 89.3 / 45.0 | 86.2 / 40.4 | 90.6 / 45.5 | 96.1 / 55.4 | **96.3 / 55.8** |

*Table 16.* Multi-class anomaly localization results on VisA using P-AUPRO metrics.

| Method → Category ↓ | | UniAD NeurIPS'22 | UniAD+Ours Ours | GLAD ECCV'24 | GLAD+Ours Ours | HVQ-Trans NeurIPS'23 | HVQ-Trans+Ours Ours | AnomalDF WACV'25 | AnomalDF+Ours Ours | Dinomaly CVPR'25 | Dinomaly+Ours Ours |
|---|---|---|---|---|---|---|---|---|---|---|---|
| Complex Structure | PCB1 | 82.5 | 89.5 | 88.3 | 82.7 | 90.4 | 88.3 | 82.8 | 77.9 | **95.2** | **95.2** |
| | PCB2 | **96.7** | 82.8 | 91.7 | 84.7 | 84.1 | 85.4 | 77.7 | 82.7 | 91.3 | 91.5 |
| | PCB3 | **96.5** | 79.3 | 94.2 | 92.2 | 79.9 | 75.6 | 79.7 | 74.5 | 94.7 | 94.8 |
| | PCB4 | 84.8 | 83.9 | **94.9** | 94.7 | 84.8 | 84.4 | 83.1 | 79.7 | 94.1 | 94.7 |
| Multiple Instances | Macaroni1 | **99.6** | 95.7 | 99.1 | 99.0 | 93.9 | 96.4 | 90.2 | 91.9 | 96.4 | 96.8 |
| | Macaroni2 | 79.5 | 89.9 | 97.2 | 98.3 | 91.9 | 94.2 | 84.8 | 97.9 | 98.6 | **98.8** |
| | Capsules | 51.1 | 74.0 | 91.8 | 91.4 | 73.2 | 61.9 | 86.1 | 87.5 | 97.1 | **97.4** |
| | Candles | 93.1 | **95.3** | 92.8 | 93.0 | 94.5 | 95.2 | 94.1 | 76.9 | 95.3 | 95.3 |
| Single Instance | Cashew | 89.5 | 87.7 | 61.1 | 75.6 | 88.8 | 90.5 | 91.3 | 69.1 | **94.3** | 93.7 |
| | Chewing gum | 80.9 | 79.8 | **92.5** | 92.0 | 77.7 | 88.5 | 85.7 | 89.4 | 88.4 | 88.4 |
| | Fryum | 62.2 | 84.2 | **96.4** | 96.1 | 84.2 | 86.9 | 85.0 | 89.4 | 93.5 | 93.7 |
| | Pipe Fryum | 86.0 | 94.1 | 98.0 | **98.3** | 93.7 | 94.5 | 94.7 | 97.8 | 95.6 | 95.5 |
| | Mean | 76.1 | 86.4 | 91.5 | 91.5 | 86.4 | 86.8 | 86.3 | 86.3 | 94.5 | **94.7** |

*Table 17.* Multi-class anomaly localization results on BTAD using I-AUROC/I-AP/I-F1max metrics.

| Method → Category ↓ | Dinomaly CVPR'25 | Dinomaly+Ours Ours | HVQ-Trans NeurIPS'23 | HVQ-Trans+Ours Ours |
|---|---|---|---|---|
| 01 | 96.8 / 98.8 / 94.9 | 97.0 / 98.9 / 95.8 | 96.9 / 98.8 / 94.9 | **98.3 / 99.4 / 96.9** |
| 02 | **89.7 / 98.4 / 93.9** | **89.7 / 98.4 / 93.9** | 75.8 / 95.9 / 92.8 | 81.7 / 97.0 / 92.8 |
| 03 | **99.9** / 98.4 / 97.6 | **99.9** / 98.4 / 97.6 | **99.9** / 98.8 / 96.8 | **99.9 / 99.6 / 98.4** |
| Mean | 95.4 / 98.5 / 95.5 | **95.5 / 98.6 / 95.8** | 90.9 / 97.8 / 94.8 | 93.3 / **98.6 / 96.0** |

*Table 18.* Multi-class anomaly localization results on BTAD using P-AUROC/P-AP/P-F1max/P-AUPRO metrics.

| Method → Category ↓ | Dinomaly CVPR'25 | Dinomaly+Ours Ours | HVQ-Trans NeurIPS'23 | HVQ-Trans+Ours Ours |
|---|---|---|---|---|
| 01 | 97.1 / 62.9 / 64.9 / 72.4 | **97.3 / 66.9 / 65.6** / 73.4 | 96.4 / 46.8 / 50.9 / 75.6 | 97.1 / 51.3 / 53.8 / **76.6** |
| 02 | 96.8 / 72.7 / 68.4 / 59.4 | **97.0 / 77.0 / 70.4 / 59.7** | 94.6 / 48.8 / 55.2 / 55.8 | 95.0 / 44.1 / 51.2 / 57.8 |
| 03 | 99.7 / 74.6 / 71.9 / 97.8 | **99.9 / 78.8 / 73.5 / 99.3** | 99.0 / 34.1 / 39.9 / 95.4 | **99.9** / 45.5 / 45.6 / 97.9 |
| Mean | 97.9 / 70.1 / 68.0 / 76.5 | **98.1 / 74.3 / 69.8 / 77.5** | 96.7 / 43.2 / 48.7 / 75.6 | 97.3 / 47.0 / 50.2 / 76.2 |

*Table 19.* Multi-class anomaly localization results on MPDD using I-AUROC/I-AP/I-F1max metrics.

| Method → | Dinomaly | Dinomaly+Ours | HVQ-Trans | HVQ-Trans+Ours |
|---|---|---|---|---|
| Category ↓ | CVPR'25 | Ours | NeurIPS'23 | Ours |
| Bracket black | 93.4 / 96.3 / 87.9 | **93.8 / 96.5** / 89.1 | 92.8 / 95.6 / **89.8** | 91.3 / 93.9 / 89.3 |
| Bracket brown | **95.3 / 96.9 / 95.3** | **95.3 / 96.9 / 95.3** | 89.4 / 93.4 / 90.3 | 93.7 / 96.2 / 94.4 |
| Bracket white | **99.0 / 99.1** / 94.7 | **99.0 / 99.1 / 94.9** | 79.2 / 82.6 / 74.7 | 92.1 / 93.0 / 85.2 |
| Connector | **100.0 / 100.0 / 100.0** | **100.0 / 100.0 / 100.0** | 89.3 / 69.3 / 81.3 | 97.9 / 96.0 / 90.3 |
| Metal plate | **100.0 / 100.0 / 100.0** | **100.0 / 100.0 / 100.0** | 97.5 / 99.1 / 95.8 | 98.6 / 99.5 / 97.3 |
| Tubes | 95.9 / 98.5 / 95.5 | **96.1 / 98.5 / 95.5** | 70.6 / 87.2 / 81.7 | 84.6 / 93.7 / 85.3 |
| Mean | 97.3 / **98.5** / 95.6 | **97.4 / 98.5 / 95.8** | 86.5 / 87.9 / 85.6 | 93.1 / 95.4 / 90.3 |

*Table 20.* Multi-class anomaly localization results on MPDD using P-AUROC/P-AP/P-F1max/P-AUPRO metrics.

| Method → | Dinomaly | Dinomaly+Ours | HVQ-Trans | HVQ-Trans+Ours |
|---|---|---|---|---|
| Category ↓ | CVPR'25 | Ours | NeurIPS'23 | Ours |
| Bracket black | **99.4 / 37.5 / 47.1 / 98.3** | **99.4** / 37.3 / 46.9 / **98.3** | 97.0 / 1.5 / 3.5 / 90.1 | 97.2 / 3.1 / 7.7 / 87.3 |
| Bracket brown | 98.2 / 50.3 / 48.5 / 96.7 | 98.2 / 50.2 / 48.8 / 96.5 | **98.3** / 31.5 / 36.7 / 88.6 | 97.3 / 31.1 / 34.4 / 63.0 |
| Bracket white | **99.4** / 18.4 / **25.1** / 93.4 | **99.4 / 19.0 / 25.1 / 93.7** | 95.2 / 0.7 / 2.6 / 84.2 | 98.0 / 6.8 / 15.5 / 88.6 |
| Connector | 99.3 / **74.6 / 69.3** / 97.5 | 99.3 / 74.5 / 69.2 / **97.6** | 97.5 / 16.6 / 27.1 / 91.1 | 97.7 / 28.8 / 33.8 / 86.9 |
| Metal plate | **99.6** / 97.7 / 92.3 / **97.7** | 99.5 / **97.7 / 92.4 / 97.7** | 96.6 / 73.6 / 74.4 / 86.6 | 98.4 / 88.1 / 82.1 / 84.6 |
| Tubes | **99.1** / 82.1 / 76.5 / **96.5** | **99.1 / 82.3 / 76.7 / 96.5** | 96.7 / 34.4 / 38.5 / 87.5 | 96.2 / 47.0 / 48.6 / 86.9 |
| Mean | 99.1 / 60.1 / 59.8 / **96.7** | **99.2 / 60.2 / 59.9 / 96.7** | 96.9 / 26.4 / 30.5 / 88.0 | 97.5 / 34.1 / 37.0 / 82.9 |

