# OpenReview forum: "CostFilter-AD: Enhancing Anomaly Detection through Matching Cost Filtering"
_ICML.cc/2025/Conference — ICML 2025 poster_

### Official Review · Reviewer_LUDM · 2025-03-11

**Overall Recommendation:** 2

**Summary:**

This paper presents a novel approach to unsupervised anomaly detection (UAD) called CostFilter-AD. Unlike traditional methods that suffer from inaccurate matching processes, this approach leverages cost volume filtering, a technique borrowed from depth and flow estimation tasks, to enhance detection accuracy. By constructing a matching cost volume and employing a filtering network, CostFilter-AD refines feature matching between input images and normal samples to effectively suppress noise while preserving critical edge information. It serves as a versatile post-processing plug-in that can be integrated with both reconstruction-based and embedding-based UAD methods. Extensive experiments demonstrate the superiority of CostFilter-AD in achieving state-of-the-art performance on multi-class and single-class UAD tasks.

**Claims And Evidence:**

The strategy proposed by the author is simple and effective, and has been fully proved.

**Essential References Not Discussed:**

None

**Experimental Designs Or Analyses:**

The experimental design and analysis are reasonable. The use of MVTec-AD and VisA datasets to validate both single-class and multi-class anomaly detection effectively demonstrates the method's strong generalization capability.

**Methods And Evaluation Criteria:**

The experimental metrics adopted by the author are based on other experiments.
 If the comparison of the calculation cost and memory cost introduced by the current module can be added, the experiment will be more complete.

**Other Comments Or Suggestions:**

None

**Other Strengths And Weaknesses:**

My biggest concern is the performance of the CostFilter-AD still shows some gaps compared to SOTA methods, and even it introduces some performance decrease.

**Questions For Authors:**

See above.

**Relation To Broader Scientific Literature:**

The paper introduces the concept of matching cost filtering, which has been used in stereo matching (e.g, optical flow estimation) but has not been used in UAD.

**Theoretical Claims:**

Although the method lacks mathematical proofs, its experiments and ablation studies investigate the existence and impact of matching noise.

---

> ### Author Rebuttal · Authors · 2025-03-31
>
> **Q1:** The proposed method is simple, effective, and well-proven. Calculation cost and memory cost can be added.
>
> **A1:** Thank you for acknowledging our work's effectiveness and suggesting the inclusion of computational and memory costs. We include comparisons of parameter count, FLOPs, memory usage, inference time, and overall training time in Table R6, showing that CostFilter-AD introduces marginal overhead while consistently improving AD performance (Table R7).
>
> **Table R6**. Comparison of computational and memory costs. ''-'' denotes training-free.
> |Method|#Params|FLOPs|Memory(GB)|Inference time(s/image)|Train time(h)|
> |:--:|:--:|:--:|:--:|:--:|:--:|
> |UniAD/+Ours|7.7M/+43.0M|198.0G/207.8G|4.53/+0.56|0.01/+0.04|14.78/+1.36|
> |Glad/+Ours|1.3B/+43.8M|>2.2T/261.3G|8.79/+2.07|3.96/+0.37|10.07/+4.95|
> |HVQ-Trans/+Ours|18.0M/+43.0M|7.4G/207.8G|4.78/+0.94|0.05/+0.07|21.79/+5.18|
> |AnomalDF/+Ours|21.0M/+43.8M|4.9G/261.3G|3.25/+0.82|0.31/+0.32|-/+17.31|
> |Dinomaly/+Ours|132.8M/+43.6M|104.7G/114.6G|4.32/+1.11|0.11/+0.05|2.31/5.49|
>
> **Table R7**. Experimental comparison with baselines and +Ours.
> | Dataset|Method|I-AUROC|I-AP|I-F1-max|P-AUROC|P-AP|P-F1-max|P-AUPRO|
> |:--:|:--:|:--:|:--:|:--:|:--:|:--:|:--:|:--:|
> |**MVTec-AD**|UniAD|97.5|99.1|97|96.9|44.5|50.5|90.6|
> | |UniAD+Ours|99|99.7|98.1|97.5|60.5|59.9|91.3|
> | |Glad|97.5|98.8|96.8|97.3|58.8|59.7|92.8|
> | |Glad+Ours|98.7|99.6|97.8|98.2|66.8|64.4|94.1|
> | |HVQ-Trans|97.9|99.3|97.4|97.4|49.4|54.3|91.5|
> | |HVQ-Trans+Ours|99|99.7|98.6|97.9|58.1|61.2|93.2|
> | |AnomalDF|96.8|98.6|97.1|98.1|61.3|60.8|93.6|
> | |AnomalDF+Ours|98.5|99.4|97.8|98.8|67.8|64.9|94.1|
> | |Dinomaly|99.6|99.8|99|98.3|68.7|68.7|94.6|
> | |Dinomaly+Ours|99.7|99.8|99.1|98.44|68.9|68.9|94.8|
> |**VisA**|UniAD|91.5|93.6|88.5|98|32.7|38.4|76.1|
> | |UniAD+Ours|92.1|94|88.9|98.6|34|39|86.4|
> | |Glad|90.1|91.4|86.7|97.4|33.9|39.4|91.5|
> | |Glad+Ours|93.2|94.1|89.2|98.1|40.7|43.7|91.5|
> | |HVQ-Trans|91.5|93.4|88.1|98.5|35.5|39.6|86.4|
> | |HVQ-Trans+Ours|93.4|95.2|89.3|98.6|41.4|45|86.8|
> | |AnomalDF|90.5|91.4|86.2|97.4|39.6|40.4|86.3|
> | |AnomalDF+Ours|94.3|95.1|90.6|99.2|44.6|45.5|86.3|
> | |Dinomaly|98.7|98.9|96.1|98.7|52.5|55.4|94.5|
> | |Dinomaly+Ours|98.7|99|96.3|98.8|53.2|55.8|94.7|
> |**MPDD**|HVQ-Trans|86.5|87.9|85.6|96.9|26.4|30.5|88.0|
> ||HVQ-Trans+Ours|93.1|95.4|90.3|97.5|34.1|37.0|82.9|
> | |Dinomaly|97.3|98.5|95.6|99.1|60|59.8|96.7|
> | |Dinomaly+Ours|97.5|98.5|95.8|99.2|60.2|59.9|96.7|
> |**BTAD**|HVQ-Trans|90.9|97.8|94.8|96.7|43.2|48.7|75.6|
> | |HVQ-Trans+Ours|93.3|98.6|96|97.3|47|50.2|76.2|
> | |Dinomaly|95.4|98.5|95.5|97.9|70.1|68|76.5|
> | |Dinomaly+Ours|95.5|98.6|95.8|98.1|74.3|69.8|77.5|
>
> **Q2:** Concern over the performance gaps compared to SOTA methods and some performance decrease.
>
> **A2:** Thank you for raising this concern.
>
> The performance gains of our method is crystal clear. To address the confusion, we would like to provide the following clarifications:
>
> 1. **Fair experimental setup**: All comparisons (baseline/+Ours) are conducted with identical image resolutions and template counts to ensure fair evaluation.
>
> 2. **Comprehensive validation**: As shown in Table R7, CostFilter-AD was applied to a range of recent multi-class anomaly detection methods, including UniAD (NeurIPS’22), GLAD (ECCV’24), HVQ-Trans (NeurIPS’23), AnomalDF (WACV’25), and Dinomaly (CVPR’25). We evaluated these models on standard benchmarks such as MVTec-AD, VisA, MPDD, and BTAD. **Consistent improvements** were observed in category-averaged metrics, see the link (https://anonymous.4open.science/r/ICML-ID8276/PDF.pdf) for more details.
>
> 3. **SOTA performance**: When integrated with Dinomaly, CostFilter-AD achieves the best performance across all evaluation metrics on the four benchmarks. For example, AUROC scores (image/pixel) reach 99.7%/98.4% on MVTec-AD, 98.7%/98.8% on VisA, 97.5%/99.2% on MPDD, and 95.5%/98.1% on BTAD.
>
> 4. **Clarifying the performance gap**: The gap you mentioned arise from different settings in **image resolution** and **number of templates**, particularly between AnomalDF (+Ours) and AnomalyDino, also noted by Reviewer kyJ7. This discrepancy is due to our method using a lightweight configuration (224×224 resolution, 3 template images per test sample), whereas AnomalDF uses higher resolutions (448 or 672) and utilizes all training images as templates. Further details are in responses A2 and A3 to Reviewer kyJ7.
>
> 5. **Category performance VS average performance**: While there may be minor fluctuations in performance within certain categories, our multi-class AD model substantially improves average metrics across datasets.
>
> 6. **Strong qualitative support**: In our submission, we provide comprehensive qualitative results in Figures 1, 3, 5, 8, 9, 10, and 11, clearly demonstrating the effectiveness and adaptability of our method.
>
> We sincerely hope that our clarifications can address your concerns. If have any remaining questions, please let us know, and we will do our best to address them.

---

### Official Review · Reviewer_fSfi · 2025-03-11

**Overall Recommendation:** 3

**Summary:**

The paper proposes CostFilter-AD, a novel method for unsupervised anomaly detection (UAD) that leverages cost volume filtering. The approach addresses matching noise issues in existing UAD methods by constructing an anomaly cost volume and refining it with a filtering network.

## Update after rebuttal

The authors addressed most of my concerns. I am just still a little worried about the computation overhead when the number of templets is large (e.g. AnomalyDINO fullshot). In addition, the improvement is relatively small when integrated on stronger backbones, e.g. AnomalyDINO-fullshot and Dinomaly

I raise my score to weak accept.

**Claims And Evidence:**

No claims apart from the priority of performance.

**Essential References Not Discussed:**

No

**Experimental Designs Or Analyses:**

The experimental setting follows the convention of UAD.

**Methods And Evaluation Criteria:**

The proposed method makes sense.

**Other Comments Or Suggestions:**

Please see Weaknesses

**Other Strengths And Weaknesses:**

Strengths:

CostFilter-AD is designed as a general post-processing module, increasing its adaptability to different anomaly detection frameworks.

Weaknesses and Questions:

1. The reported performance of AnomalDF (AnomalyDINO) is notably inconsistent with published results. In its original paper, AnomalyDINO (essentially DINOv2+PatchCore) achieved 99.3/97.2 image-AUROC on MVTec-AD and Visa, respectively. However, this paper reports substantially lower scores of 96.8/90.5. Even with CostFilter-AD ensembling, the results only reach 98.5/94.3, still below the original benchmarks.

Upon closer examination of the Appendix, I discovered the authors limited AnomalyDINO to merely 3 shots as normal supports, citing "substantial storage overhead" concerns for full-shot approaches. So it is actually few-shot?

Moreover, the storage overhead claim appears questionable. Memory-bank methods like PatchCore and AnomalyDINO can operate effectively on standard laptops using the "faiss" package. The memory-bank storage requirements are comparable to the model size itself. Additionally, PatchCore already introduced coreset-subsampling specifically to reduce storage requirements and optimize nearest-neighbor search efficiency.

I would recommend that the authors evaluate CostFilter-AD ensembled with full-shot implementations of both AnomalyDINO and PatchCore for a more accurate and fair comparison.

2. HVQ-Trans is a feature-reconstruction method that reconstructs features of EfficientNet instead of original images. How to leverage the generated images from it?

3. Following the above, can CostFilter-AD be ensembled on feature-reconstruction methods, such as RD4AD, UniAD, MambaAD, Dinomaly, etc.? Feature-reconstruction-based methods are the most popular branch in multi-class UAD.

4. More datasets are suggested, such as Real-AD, BTAD, MPDD, etc.

**Questions For Authors:**

Please see Weaknesses

**Relation To Broader Scientific Literature:**

In my assessment, this represents a framework similar to DRAEM (or DesTSeg), comprising two primary components: a normal-restoration network (analogous to the autoencoder in DRAEM) and a supervised segmentation network that segments anomalous based on two input images..

**Theoretical Claims:**

None

---

> ### Author Rebuttal · Authors · 2025-03-31
>
> **Table R5**. AnomalDF (abbr. as ADF) /+Ours Comparison under a fair setting.
> |ID|Dataset|Method|Input size|#Templates|I-AUROC|I-AP|I-F1-max|P-AUROC|P-AP|P-F1-max|P-AUPRO|
> |:--:|:--:|:--:|:--:|:--:|:--:|:--:|:--:|:--:|:--:|:--:|:--:|
> |1|**MVTec-AD**|ADF|256|3|96.8|98.6|97.1|98.1|61.3|60.8|93.6|
> |2||+Ours|256|3|98.5|99.4|97.8|98.8|67.8|64.9|94.2|
> |3||ADF|256|Full|99.0|99.3|98.4|97.5|-|58.7|91.7|
> |4||+Ours|256|Full|99.3|99.8|98.6|98.9|68.7|65.5|96.6|
> |5||ADF|448|Full|99.3|99.7|98.8|97.9|-|61.8|92.9|
> |6||+Ours|448|Full|99.5|99.8|98.9|99.0|72.4|68.4|95.4|
> |7||ADF|672|Full|99.5|99.8|99.0|98.2|-|64.3|95.0|
> |8||+Ours|672|Full|99.6|99.9|99.0|99.1|74.4|69.7|96.3|
> |9|**VisA**|ADF|256|3|90.5|91.4|86.2|97.4|39.6|40.4|86.3|
> |10||+Ours|256|3|94.3|95.1|90.6|99.2|44.6|45.5|84.5|
> |11||ADF|256|Full|94.6|95.7|90.9|98.3|-|44.3|86.7|
> |12||+Ours|256|Full|95.5|96.3|91.5|99.4|45.9|46.6|87.0|
> |13||ADF|448|Full|97.2|97.6|93.7|98.7|-|50.5|95.0|
> |14||+Ours|448|Full|97.4|97.7|93.8|99.4|42.2|53.6|95.2|
> |15||ADF|672|Full|97.6|97.2|94.3|98.9|-|53.8|96.1|
> |16||+Ours|672|Full|97.8|98.0|94.6|99.4|47.6|54.5|96.4|
>
> **Q1**: Discrepancy between AnomalDF (AnomalyDINO) performance and reported results.
>
> **A1**: Thank you! The gap is primarily due to two factors:
>
> - **Number of templates**: In our experiments, we re-run AnomalyDINO using **3 randomly sampled images** from the training dataset as reference templates during testing. In contrast, the original full-shot setting of AnomalyDINO employed **the entire training dataset** as templates.
> - **Image resolution**: We resize images to **256×256**, whereas the original AnomalyDINO resized to **448×448**. Higher resolution allows the model to capture more details.
>
> As shown in the Table R5, **under AnomalyDINO's full-shot setting**, we achieve image AUROC scores of 99.5 (MVTec-AD) and 97.4 (VisA), outperforming AnomalDF's 99.3/97.2.
>
> **Q2**: Clarification on the 3-shot setting: Is the AnomalDF+Ours actually few-shot?
>
> **A2**: The answer is No. The AnomalDF (+Ours) in our paper is full-shot, but differs from the original AnomalyDINO setting.
>
> - Training: AnomalDF (+Ours) uses a fixed number (N=3) of templates per input image, randomly sampled from the full training set for each input, rather than drawn from a fixed template set as in the original few-shot setting of AnomalyDINO. Our dynamic sampling ensures the template pool covers the full training distribution; thus, we classify it as full-shot, offering a trade-off between template diversity and memory efficiency.
> - Test: For fairness, we evaluate AnomalDF (+Ours) using our dynamic 3-shot sampling protocol, as reflected in the results reported in our submission.
>
> **Q3**: Storage overhead claim.
>
> **A3**: Thanks for your kindly reminding. We will carefully revise our statement on “storage overhead” to provide a more accurate and precise explanation.
>
> **Q4**: Evaluation of full-shot AnomalyDINO.
>
> **A4**: Following your advice, we tested under the original full-shot AnomalyDINO setting. In Table R5, Exp. 3–8 report results on MVTec-AD and Exp. 10–16 on VisA. CostFilter-AD **consistently improves** AnomalDF’s performance across all resolutions, with some cases (e.g., Exp. 6 vs. 7) showing that AnomalDF+Ours at lower resolutions matches or surpasses the baseline at higher resolutions.
>
> We are integrating CostFilter-AD with PatchCore, similar to AnomalDF (+Ours), and will report it in the revised version.
>
> **Q5**: How to utilize the reconstructed features from HVQ-Trans.
>
> **A5**: Thanks! We directly use the input and reconstructed features from HVQ-Trans to construct the cost volume, without decoding them back to the image domain, as the HVQ-Trans already provides the necessary feature representations. This differs from Glad+Ours, which uses a pre-trained encoder to extract image features. We will state the implementation of HVQ-Trans+Ours more clearly in the revised vision.
>
> **Q6**: Following the above, can CostFilter-AD be ensembled on feature-reconstruction methods like RD4AD, UniAD, MambaAD, Dinomaly?
>
> **A6**: Yes! Similar to HVQ-Trans (+Ours), for feature-reconstruction-based methods, we bypass the feature encoder and directly construct the cost volume using both input and reconstructed features. We further validated it by integrating with UniAD and Dinomaly on MVTec-AD and VisA. SOTA results show CostFilter-AD's generalizability (see Table R7 below and the link (https://anonymous.4open.science/r/ICML-ID8276/PDF.pdf) for details).
>
> **Q7**: More datasets.
>
> **A7**: Thanks! We have extended our evaluation by integrating CostFilter-AD with HVQ-Trans and Dinomaly on two more datasets: BTAD and MPDD. As shown in Table R7 and the link (https://anonymous.4open.science/r/ICML-ID8276/PDF.pdf), we consistently improve baseline performance, validating effectiveness.
>
> We sincerely appreciate your feedback and hope our response clarifies your concerns. If you have further questions, please feel free to let us know, and we'll be glad to clarify.

---

### Official Review · Reviewer_kyJ7 · 2025-03-13

**Overall Recommendation:** 3

**Summary:**

This paper introduces cost filtering into unsupervised anomaly detection and multi-class anomaly detection. The authors offer a new perspective to differentiate the discrepancy between the input and templates. Their experiments appear to demonstrate the effectiveness of the proposed method.

**Claims And Evidence:**

Yes

**Essential References Not Discussed:**

No

**Experimental Designs Or Analyses:**

Yes

**Methods And Evaluation Criteria:**

Yes

**Other Comments Or Suggestions:**

No

**Other Strengths And Weaknesses:**

Strengths:

1. This paper incorporates cost filtering into unsupervised anomaly detection, providing an interesting perspective that supplements the current literature on anomaly detection.

2. The paper is well-organized with effective use of visual aids, allowing readers to follow the writing flow smoothly.

3. The authors conduct extensive experiments to illustrate the effectiveness of their proposed method. The quantitative results highlight the method's effectiveness.

Weaknesses:

1. It is unclear why the authors neglect the first multi-class anomaly detection work, UniAD, and do not provide a performance comparison with it.

2. The paper introduces a cost filtering network to model the discrepancy between input samples and templates, which may increase memory usage and computational overhead. Besides detection performance, parameter efficiency is also an important metric for evaluating an algorithm. MOE-AD [1] is a recent multi-class detection framework that emphasizes parameter efficiency. The authors should supplement their experiments to demonstrate the computational efficiency and effectiveness of their method.

3. It would be beneficial if the authors could provide failure cases to enable a deeper analysis of the proposed method. This addition would offer valuable insights for the readers.

[1] Meng, S., Meng, W., Zhou, Q., Li, S., Hou, W., & He, S. (2024). MoEAD: A Parameter-Efficient Model for Multi-class Anomaly Detection. European Conference on Computer Vision.

**Questions For Authors:**

See Weaknesses

**Relation To Broader Scientific Literature:**

This paper proposed a new approach to solve unsupervised anomaly detection.

**Theoretical Claims:**

Yes

---

> ### Author Rebuttal · Authors · 2025-03-31
>
> **Q1**: Missing the first multi-class anomaly detection work, UniAD, and a performance comparison.
>
> **A1**: Thanks for your reminding.
>
> - We fully recognize UniAD (NeurIPS'22) as a pioneering work in multi-class anomaly detection. In response, we have conducted extensive evaluations by integrating CostFilter-AD into UniAD, and will include both citation and performance comparisons in the revised paper.
> - As shown in Table R3, on MVTec-AD, this integration improves image-level AUROC/AUPRC/F1-max by +1.5\%/+0.6\%/+1.1\%, and pixel-level AUROC/AUPRC/F1-max/AUPRO by +0.6\%/+16\%/+9.4\%/+0.7\%, respectively. On VisA, we observe gains of +0.6\%/+0.4\%/+0.4\% at the image level and +0.6\%/+1.3\%/+0.6\%/+10.3\% at the pixel level.
> - Furthermore, we incorporated CostFilter-AD into Dinomaly (CVPR’25) and validated its effectiveness across more benchmarks. These consistent gains further highlight the effectiveness, flexibility, and generalizability of our method. Please refer to the link (https://anonymous.4open.science/r/ICML-ID8276/PDF.pdf) for more category-aware details.
>
> **Table R3**. Integration of our CostFilter-AD with UniAD and Dinomaly baselines.
> |Dataset|Method|I-AUROC|I-AP|I-F1-max|P-AUROC|P-AP|P-F1-max|P-AUPRO|
> |:--:|:--:|:--:|:--:|:--:|:--:|:--:|:--:|:--:|
> |**MVTec-AD**|UniAD|97.5|99.1|97|96.9|44.5|50.5|90.6|
> ||UniAD+Ours|99|99.7|98.1|97.5|60.5|59.9|91.3|
> ||Dinomaly|99.6|99.8|99|98.3|68.7|68.7|94.6|
> ||Dinomaly+Ours|99.7|99.8|99.1|98.4|68.9|68.9|94.8|
> |**VisA**|UniAD|91.5|93.6|88.5|98|32.7|38.4|76.1|
> ||UniAD+Ours|92.1|94|88.9|98.6|34|39|86.4|
> ||Dinomaly|98.7|98.9|96.1|98.7|52.5|55.4|94.5|
> ||Dinomaly+Ours|98.7|99|96.3|98.8|53.2|55.8|94.7|
>
> **Q2**: Besides detection performance, parameter efficiency is crucial. MOE-AD highlights this in multi-class detection. The authors should provide results to demonstrate the efficiency and effectiveness.
>
> **A2**: Thank you for highlighting the parameter efficiency.
>
> - MoE-AD presents an elegant solution by significantly reducing model size via recursive ViT blocks and MoE-based FFN selection, setting a strong benchmark for balancing accuracy and efficiency in resource-constrained scenarios. We will cite this excellent work in our revised paper and are excited to explore its potential synergy with CostFilter-AD in future work.
>
> - In response, we present a detailed comparison across multiple baselines and benchmarks (see Table R4), reporting parameters, FLOPs, memory usage, and inference time. In terms of **memory usage**, the increase is negligible. Regarding **computational overhead**, it remains relatively low compared to diffusion-based methods. For other approaches, the overhead can be further minimized by converting global matching into local matching, thereby optimizing the cost volume for more efficient computation. Regarding **efficiency**, our method provides notable performance gains with a reasonable increase in inference time.
>
> **Table R4**. Parameter efficiency comparison and performance gain of baselines/+Ours on MVTec-AD.
> |Method|#Params|FLOPs|Memory(GB)|Inference time (s/image)|I-AUROC|P-AUROC|
> |:--:|:--:|:--:|:--:|:--:|:--:|:--:|
> |UniAD/+Ours|7.7M/+43.0M|198.0G/207.8G|4.53/+0.56|0.01/+0.04|97.5/+1.5|96.9/+0.6|
> |Glad/+Ours|1.3B/+43.8M|>2.2T/261.3G|8.79/+2.07|3.96/+0.37|97.5/+1.2|97.3/+0.9|
> |HVQ-Trans/+Ours|18.0M/+43.0M|7.4G/207.8G|4.78/+0.94|0.05/+0.07|97.9/+1.1|97.4/+0.5|
> |AnomalDF/+Ours|21.0M/+43.8M|4.9G/261.3G|3.25/+0.82|0.31/+0.32|96.8/+1.7|98.1/+0.7|
> |Dinomaly/+Ours|132.8M/+43.6M|104.7G/114.6G|4.32/+1.11|0.11/+0.05|99.6/+0.1|98.3/+0.9|
>
> Notably, the #Params in our model varies slightly across different baselines. This is because we need to map the matching features, which have different #Channels (e.g., 196, 768, or 1024), into a unified 96-dimensional space.
>
> **Q3**: It would be beneficial if the authors could provide failure cases to enable a deeper analysis.
>
> **A3**: Thank you for the insightful suggestion. As illustrated in Fig. 1 (at link (https://anonymous.4open.science/r/ICML-ID8276/PDF.pdf)), we present representative failure cases from six categories in MVTec-AD and VisA, demonstrating the performance of our method before and after filtering.
>
> While our approach effectively reduces matching noise, its effectiveness depends on the presence of anomaly-relevant signals in the cost volume. If these signals are absent—due to low-resolution inputs or insufficient feature learning—the filtering process cannot recover them. In other words, as a denoising rather than a generative module, CostFilter-AD enhances existing features but cannot infer anomalies from missing evidence.
>
> We appreciate your feedback and hope that our revisions address your concerns. If there are any remaining issues or questions, please let us know, and we will respond promptly.

---

### Official Review · Reviewer_Yvmv · 2025-03-13

**Overall Recommendation:** 3

**Summary:**

The paper introduces the concept of cost volume filtering, combined ideas from stereo matching and depth estimation, into the field of unsupervised anomaly detection. This method addresses the often-overlooked matching noise issue, which is a common challenge in existing AD methods.

**Claims And Evidence:**

The paper is supported by solid claims and motivations. The paper provides extensive quantitative results on two benchmark datasets (MVTec-AD and VisA), and these results are complemented by qualitative visualizations and thorough ablation studies. However, the proposed method plug-in design can be applied in multiple different AD baselines. It could be better if the authors could apply this into multiple AD baselines.

**Essential References Not Discussed:**

Some of the SOTA methods have been missing in the paper.

Lee, Mingyu, and Jongwon Choi. "Text-guided variational image generation for industrial anomaly detection and segmentation." Proceedings of the IEEE/CVF Conference on Computer Vision and Pattern Recognition. 2024.

Chen, Yuanhong, et al. "Deep one-class classification via interpolated gaussian descriptor." Proceedings of the AAAI Conference on Artificial Intelligence. Vol. 36. No. 1. 2022.

Bae, Jaehyeok, Jae-Han Lee, and Seyun Kim. "Pni: industrial anomaly detection using position and neighborhood information." Proceedings of the IEEE/CVF International Conference on Computer Vision. 2023.

**Experimental Designs Or Analyses:**

The experimental design in the paper is generally sound and well thought out
The experiments are conducted on established datasets (MVTec-AD and VisA) which are widely recognized for anomaly detection.
The paper evaluates both multi-class and single-class anomaly detection scenarios, and the integration of the proposed CostFilter-AD as a plug-in for different base models (reconstruction-based and embedding-based) demonstrates its generality and robustness.

**Methods And Evaluation Criteria:**

The proposed methods and evaluation criteria are well aligned with the problem of unsupervised anomaly detection.

However, the paper mainly reports results using AUC metrics for both image-level and pixel-level anomaly detection. It would be better if the authors also report metric such as PR, or AUPRC, where many other AD papers have benchmarked their methods using this metric given that AUC results are very high and close to saturation and PR curves are often more sensitive to changes in the precision-recall trade-off, particularly in imbalanced datasets where false positives or negatives have different impacts on precision and recall.

**Other Comments Or Suggestions:**

NA

**Other Strengths And Weaknesses:**

the overall pipeline is quite complex. The multiple steps—from multi-layer feature extraction, cost volume construction, and 3D U-Net filtering, to dual-stream attention and class-aware adaptation—may increase the difficulty of implementation. Could the authors provide how the model sensitive to architectures or hyper-parameters selections. Also it would be valuable for authors to compute the complexity like inference/training time?

**Questions For Authors:**

Please see above for questions and concerns.

**Relation To Broader Scientific Literature:**

Overall the paper present a decent and novel plug and play module for anomaly detection and show great performances. This can relates to broader impact in the field of anomaly detection.

**Theoretical Claims:**

The paper primarily focuses on empirical validation and algorithmic design rather than on formal theoretical proofs.  There are no formal theorems or rigorous proofs provided to mathematically guarantee the properties of the proposed method. The authors rely on intuitive reasoning and extensive experimental validation to support their claims.

---

> ### Author Rebuttal · Authors · 2025-03-31
>
> **Q1**: The paper is supported by solid claims and motivations. It could be better if the authors could apply the proposed method plug-in design into multiple AD baselines.
>
> **A1**: Thanks! We apply our method to **UniAD** (NeurIPS'22) (**new add**), **GLAD** (ECCV'24), **HVQ-Trans** (NeurIPS'23), **AnomalDF** (WACV'25), **Dinomaly** [r1] (CVPR'25) (**new add**) on benchmarks **MVTec-AD**, **VisA**, **BTAD** (**new add**), and **MPDD** (**new add**). Please refer to Table R1 for category-averaged metrics, with detailed per-category metrics available in the link (https://anonymous.4open.science/r/ICML-ID8276/PDF.pdf).
>
> [r1] Jia Guo, et al. Dinomaly: The less is more philosophy in multi-class unsupervised anomaly detection. CVPR 2025.
>
> **Table R1**. Evaluation with more baselines on multiple benchmarks via 7 comprehensive metrics.
> |Dataset|Method|I-AUROC|I-AP|I-F1-max|P-AUROC|P-AP|P-F1-max|P-AUPRO|
> |:--:|:--:|:--:|:--:|:--:|:--:|:--:|:--:|:--:|
> |**MVTec-AD**|UniAD|97.5|99.1|97|96.9|44.5|50.5|90.6|
> ||UniAD+Ours|99|99.7|98.1|97.5|60.5|59.9|91.3|
> ||Dinomaly|99.6|99.8|99|98.3|68.7|68.7|94.6|
> ||Dinomaly+Ours|99.7|99.8|99.1|98.4|68.9|68.9|94.8|
> |**VisA**|UniAD|91.5|93.6|88.5|98|32.7|38.4|76.1|
> ||UniAD+Ours|92.1|94|88.9|98.6|34|39|86.4|
> ||Dinomaly|98.7|98.9|96.1|98.7|52.5|55.4|94.5|
> ||Dinomaly+Ours|98.7|99|96.3|98.8|53.2|55.8|94.7|
> |**MPDD**|HVQ-Trans|86.5|87.9|85.6|96.9|26.4|30.5|88.0|
> ||HVQ-Trans+Ours|93.1|95.4|90.3|97.5|34.1|37.0|82.9|
> ||Dinomaly|97.3|98.5|95.6|99.1|60|59.8|96.7|
> ||Dinomaly+Ours|97.5|98.5|95.8|99.2|60.2|59.9|96.7|
> |**BTAD**|HVQ-Trans|90.9|97.8|94.8|96.7|43.2|48.7|75.6|
> ||HVQ-Trans+Ours|93.3|98.6|96|97.3|47|50.2|76.2|
> ||Dinomaly|95.4|98.5|95.5|97.9|70.1|68|76.5|
> ||Dinomaly+Ours|95.5|98.6|95.8|98.1|74.3|69.8|77.5|
>
> **Q2**: It would be better if the authors also report metric such as PR, or AUPRC.
>
> **A2**: Thank you! We would like to clarify that AUPRC (i.e., the area under the precision-recall curve) has already been reported in Tables 7–10 of our submission, where we refer to it as I-AP (image-level AUPRC) and P-AP (pixel-level AUPRC), following the terminology adopted by Glad, AnomalDF, and Dinomaly. The AUPRC is computed by the **`average_precision_score`** function from **`sklearn.metrics`**. We will clarify this in the revised version to avoid confusion.
>
> **Q3**: Some of the SOTA methods have been missing in the paper.
>
> **A3**: Thank you! We acknowledge the importance of the SOTA methods you mentioned and will incorporate both citations and discussions in the revised version for a more thorough and balanced evaluation.
>
> **Q4**: The multiple steps may increase the difficulty of implementation. Could the authors provide how the model sensitive to architectures or hyper parameters selections.
>
> **A4**: Thanks!
> - To address potential concerns about implementation and reproducibility, we will release the full source code and model weights.
>
> - We have also conducted comprehensive ablation studies in Table 4 to examine the model's sensitivity to architectural choices. Specifically, we evaluate different cost volume constructions (DN→depth/channel), template selection strategies ($C_0$ vs. $C_{N-1}$), dual-stream attention (SG and MG), and class-aware adaptation (loss $\mathcal{L}_s$). Results confirm that each component plays a vital role in achieving strong performance, validating their necessity and effectiveness.
>
> **Q5**: Also it would be valuable for authors to compute the complexity like inference/training time.
>
> **A5**: Thanks! We have provided the inference time and memory usage in Table 6 of the submission. These metrics, along with overall training time, are provided in Table R2 below. As shown, our method incurs reasonable computational overhead and minimal memory usage, while delivering notable performance improvements.
>
> **Table R2**. Complexity comparison of baselines and +Ours. ''-'' denotes training-free.
> |Method|Inference time (s/image)|Train time (h)|Memory (GB)|
> |:--:|:--:|:--:|:--:|
> |UniAD/+Ours|0.01/+0.04|14.78/+1.36|4.53/+0.56|
> |Glad/+Ours|3.96/+0.37|10.07/+4.95|8.79/+2.07|
> |HVQ-Trans/+Ours|0.05/+0.07|21.79/+5.18|4.78/+0.94|
> |AnomalDF/+Ours|0.31/+0.32|-/+17.31|3.25/+0.82|
> |Dinomaly/+Ours|0.11/+0.05|2.31/5.49|4.32/+1.11|
>
> We sincerely appreciate your feedback and hope our response has addressed your concerns. If you have any additional questions, please let us know, and we will respond promptly.

---

### Decision · Program_Chairs · 2025-05-01

**Decision:**

Accept (poster)

**Comment:**

This paper received mixed scores, including three weak accepts and one weak reject. Most reviewers acknowledged that the proposed method is well-designed and that the experiments are thorough. Initially, reviewers raised several concerns, including the need for comparisons with additional baselines (Reviewers Yvmv and fSfi), validation on more datasets (Reviewer fSfi), computational efficiency (Reviewers Yvmv, kyJ7, and LUDM), fairness in comparisons (Reviewer fSfi), and the extent of performance gains (Reviewer LUDM). However, all of these concerns were fully addressed during the rebuttal. One reviewer noted that the performance still lags slightly behind some state-of-the-art methods. Nevertheless, they acknowledged that the performance could be further improved with more advanced backbone models, and that the contribution remains novel and significant. AC has also reviewed the paper, the reviewer comments, and the rebuttal, and agrees that the motivation is clear, the paper is well-written, and the experimental evaluation is solid. Therefore, the AC recommends acceptance. It is encouraged that all additional experiments and discussions presented in the rebuttal be incorporated into the final version.